# End-to-End Stochastic Optimization with Energy-Based Model

**Lingkai Kong**   **Jiaming Cui**   **Yuchen Zhuang**   **Rui Feng**
**B. Aditya Prakash**   **Chao Zhang**
College of Computing
Georgia Institute of Technology
`{lkkong,jiamingcui1997,yczhuang,rfeng,badityap,chaozhang}@gatech.edu`

## Abstract

Decision-focused learning (DFL) was recently proposed for stochastic optimization problems that involve unknown parameters. By integrating predictive modeling with an implicitly differentiable optimization layer, DFL has shown superior performance to the standard two-stage predict-then-optimize pipeline. However, most existing DFL methods are only applicable to convex problems or a subset of non-convex problems that can be easily relaxed to convex ones. Further, they can be inefficient in training due to the requirement of solving and differentiating through the optimization problem in every training iteration. We propose SO-EBM, a general and efficient DFL method for stochastic optimization using energy-based models. Instead of relying on KKT conditions to induce an implicit optimization layer, SO-EBM explicitly parameterizes the original optimization problem using a differentiable optimization layer based on energy functions. To better approximate the optimization landscape, we propose a coupled training objective that uses a maximum likelihood loss to capture the optimum location and a distribution-based regularizer to capture the overall energy landscape. Finally, we propose an efficient training procedure for SO-EBM with a self-normalized importance sampler based on a Gaussian mixture proposal. We evaluate SO-EBM in three applications: power scheduling, COVID-19 resource allocation, and non-convex adversarial security game, demonstrating the effectiveness and efficiency of SO-EBM.

## 1 Introduction

Many real-life decision making tasks are stochastic optimization problems, where one needs to make decisions to minimize a cost function that involves stochastic parameters. Oftentimes, the involved stochastic parameters are *unknown* and *context-dependent*, meaning that they need to be predicted from observed features. For example, when allocating clinical resources for COVID-19, it is necessary to consider future cases in different regions, whose distributions are unknown and have to be predicted from the current state. As two other examples, hedge funds need to continuously adjust their portfolio for maximal expected return, by forecasting future return rates of different stocks; and in supply chain optimization, facility locations need to be decided to minimize long-term operational costs, by accounting for unknown and uncertain regional supplies and customer demands.

With the feasibility of training powerful deep learning predictors from large amounts of data, it is increasingly common to solve such stochastic optimization problems using a two-stage predict-*then*-optimize pipeline. In the prediction stage, one learns a predictive model for the unknown parameters using certain prediction loss (*e.g.*, negative likelihood). In the optimization stage, the predicted distributions of the parameters are used to parameterize the stochastic optimization problem, which can be then solved using off-the-shelf solvers [21, 2, 12]. This two-stage pipeline relies on an implicit

36th Conference on Neural Information Processing Systems (NeurIPS 2022).

and commonly-accepted assumption: *improvements in parameter prediction in terms of the prediction loss will always translate to better optimization outcomes.* However, this is not the case: machine learning models make errors and the impact of prediction errors is not uniform *w.r.t.* the optimization loss. Thus, a smaller predictive loss does not necessarily lead to a smaller decision regret.

A better approach is decision-focused learning (DFL) [13, 1, 4], which integrates prediction and optimization layers into a unified model to learn them end-to-end. Most DFL methods implement the optimization procedure as an implicitly differentiable layer and develop techniques (*e.g.*, using KKT conditions) to compute the gradients *w.r.t.* the decision variables and enable back-propagating through it. Compared to the two-stage pipeline, the prediction layer so learned is tailored for the optimization problem and better in the sense that it can yield decisions with smaller regrets. Besides DFL, there are also approaches where a policy network is trained to directly map from the input to the solution of the optimization problem using supervised or reinforcement learning [47, 26, 34]. However, their performance are often inferior to DFL as they ignore the algorithmic structure of the problem and typically require a large amount of data to rediscover the algorithmic structure.

Despite their promising results, existing DFL methods [13, 4, 1, 41, 48] suffer from two drawbacks. (1) *Generality.* To leverage the KKT condition, they are mostly applicable to only convex optimization objectives [13, 4, 1]. Though a few works approximate nonconvex objectives by quadratic functions [41, 48], their applicability is still limited to easy-to-relax nonconvex problems and can suffer from poor gradient estimates caused by the relaxation. (2) *Scalability.* Due to the reliance of using the KKT condition to compute derivatives, they require repeatedly solving the optimization program and back-propagating through it during training, which makes them unscalable. The problem is more severe when the expectation of the cost cannot be computed analytically.

We propose a new end-to-end stochastic optimization method using an energy-based model (EBM), named SO-EBM. Similar to DFL, SO-EBM models the algorithmic predict-*and*-optimize structure by stacking a differentiable optimization layer on top of a neural predictor. Different from DFL, SO-EBM eliminates the need of using KKT conditions to induce an implicit differentiable optimization layer. Instead, it leverages expressive EBMs [33] to directly model the conditional probability of the decisions and parameterizes an explicit energy-based differentiable layer as a surrogate to the original optimization problem. To better approximate the optimization landscape with the EBM surrogate, we design two complementary learning objectives in SO-EBM: 1) a local matching objective that maximizes the likelihood of the optimal decisions; and 2) a global matching objective that minimizes the distribution distance between the posterior distribution of the decision variables and the surrogate-based decision distribution.

Due to its flexibility, SO-EBM is not constrained to convex objectives, but can be applied to a wide class of stochastic optimization problems. Another key advantage of SO-EBM is its computational efficiency. As the optimization layer is parameterized by an energy-based model, SO-EBM eliminates the need of solving and differentiating through the optimization procedure at each training iteration. Rather, SO-EBM estimates the derivatives of the energy-based surrogate model using a self-normalized importance sampler. We design a Gaussian-mixture proposal distribution for the sampler, which not only reduces sampling variance, but also borrows the idea of contrastive divergence for MCMC methods to speed up the training of SO-EBM.

The main contributions of this work are: (1) We propose a new end-to-end stochastic optimization method based on energy-based model. It avoids the needs of solving and differentiating through the optimization problem during training and can be applied to a wide range of stochastic optimization problems beyond convex objectives. (2) We propose a coupled training objective to encourage the energy-based surrogate to well approximate the optimization landscape and an efficient self-normalized importance sampler based on a mixture-of-Gaussian proposal. (3) Experiments on power scheduling, COVID-19 resource allocation and adversarial network security game show that our method can achieve better or comparable performance than existing DFL methods while being much more efficient.

## 2 Preliminaries

**Problem Formulation.** Consider a stochastic optimization problem:

$$\arg\min_{a \in C} f(y, a),$$

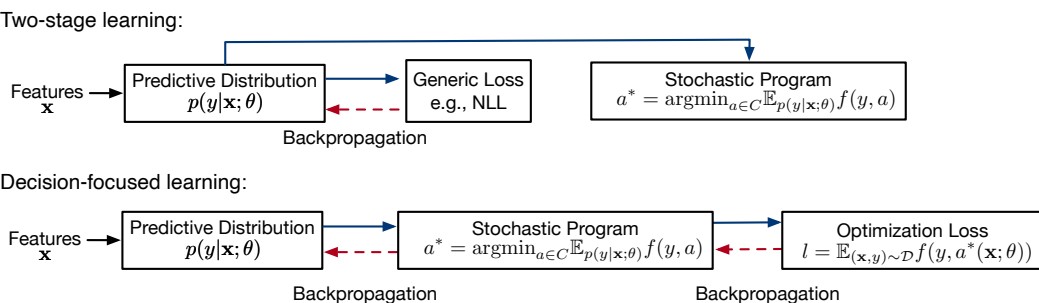

Figure 1: Two-stage learning back-propagates from a predictive loss to the model, ignoring the latter effect of the optimization problem. Decision-focused learning directly optimizes the task loss but needs to solve and back-propagate through the optimization problem at every training iteration.

where $y$ denotes the parameters of the optimization problem, $a$ denotes the decision variables within a feasible space $C$, and $f$ is the cost function to be optimized. In many applications, the parameters $y$ are *unknown* and *stochastic*, which must be inferred from some correlated features $\mathbf{x}$. We assume a dataset $\mathcal{D} = \{\mathbf{x}_i, y_i\}_{i=1}^N$ drawn from the joint distribution of the features and problem parameters. Our task is to learn a decision-making model $\mathbf{M}$ parameterized by $\theta$, which takes the features $\mathbf{x}$ as input and outputs the optimal decisions $a^*(\mathbf{x}; \theta)$. The model should be learned such that its output optimal decisions minimize the expected decision cost under the joint distribution of $(\mathbf{x}, y)$, namely:

$$\arg\min_\theta \mathbb{E}_{(\mathbf{x},y)\sim\mathcal{D}} f(y, a^*(\mathbf{x}; \theta)).$$

For example, in supply chain optimization, $y$ can be regional customer demands, $a$ can be pre-ordered products for each region, $f$ measures the gap between actual customer demands and pre-ordered supplies, and the problem is to decide the best $a$ to minimize the cost $f$. Herein, the actual customer demands $y$ are usually unknown and must be predicted from features $\mathbf{x}$ such as customer purchase history and regional economy indices.

**Two-stage learning v.s. Decision Focused Learning (DFL).** A common practice to the above stochastic optimization problem is the two-stage predict-then-optimize framework. It first learns an probabilistic predictive model $p(y|\mathbf{x}; \theta)$ and then uses existing stochastic optimization solvers to obtain the optimal action that minimizes the expected cost: $a^*(\mathbf{x}; \theta) = \arg\min_{a \in C} \mathbb{E}_{y \sim p(y|\mathbf{x};\theta)} f(y, a)$. Though the two-stage approach is simple and efficient, it can suffer from suboptimal performance due to the misalignment of the prediction loss and the optimization loss. In contrast, DFL integrates prediction and optimization into an end-to-end model, thus tailoring the predictive model for the optimization task, as shown in Figure 1. By directly optimizing the task loss, the gradient of the model parameters can be obtained with the following chain rule:

$$\frac{\partial f(y, a^*(\mathbf{x}; \theta))}{\partial \theta} = \frac{\partial f(y, a^*(\mathbf{x}; \theta))}{\partial a^*(\mathbf{x}; \theta)} \frac{\partial a^*(\mathbf{x}; \theta)}{\partial y} \frac{\partial y}{\partial \theta}.$$

The key challenge here is to compute the Jacobian $\frac{\partial a^*(\mathbf{x};\theta)}{\partial y}$: it is needed to apply the chain rule to learn the model using gradient decent methods. This is nontrivial, because $a^*$ is the solution of a stochastic optimization solver and not directly differentiable. To address this challenge, OptNet [13, 4] assumes quadratic optimization objectives and differentiates through the KKT conditions using the implicit function theorem. This way, OptNet can obtain the Jacobian by solving the optimization problem along with a set of linear equations in each training iteration. Several works [1, 41, 48] extend this technique to more general cases. For example, cvxpylayers [1] extends it to more general cases of convex optimization using disciplined parametrized programming (DPP) grammar.

Although existing DFL approaches can achieve better decisions compared to two-stage learning, they have several drawbacks. First, they are often constrained to convex optimization objectives as they rely on the KKT conditions to compute derivatives. Though [41, 48] propose to approximate some non-convex objectives by a quadratic function around a local minimum, the inaccurate gradients may be aggregated during the training iterations and thus lead to poor decision quality. Second, they suffer from high computational complexity because the computation of the Jacobian $\frac{\partial a^*(\mathbf{x};\theta)}{\partial y}$ requires repeatedly solving the optimization program and back-propagating through it. The problem is more severe when the expectation of the cost cannot be computed analytically. In that case, we need to

use sample average approximation [27, 46, 30] and draw multiple IID samples from the predictive distribution, which makes the objective much more complicated and expensive.

# 3 Proposed Method

## 3.1 Energy-based Model for End-to-end Stochastic Programming

Our task is to learn an optimal decision-making model $\mathbf{M}_\theta$, such that its output optimal decision $a^*(\mathbf{x}; \theta)$ for input $\mathbf{x}$ minimizes the expected optimization cost: $\mathbb{E}_{(\mathbf{x},y)\sim\mathcal{D}}f(y, a^*(\mathbf{x}; \theta))$. The core idea of our method is to directly model $\mathbf{M}_\theta$'s probability distribution over the decisions conditioned on the features, denoted as $q(a|\mathbf{x}; \theta)$, using energy-based parameterization [33]:

$$q(a|\mathbf{x}; \theta) = \frac{\exp(-E(\mathbf{x}, a; \theta))}{Z(\mathbf{x}, \theta)}, \quad Z(\mathbf{x}, \theta) = \int \exp(-E(\mathbf{x}, a; \theta)da. \tag{1}$$

To parameterize the energy function $E(\mathbf{x}, a; \theta)$, a natural option is to directly use a deep neural network which takes a feature-decision pair $(\mathbf{x}, a)$ as input and output a scalar value. However, such a design ignores the algorithmic structure of the optimization problem and thus can be data-inefficient during learning. Instead, we propose to explicitly model the problem structure by using the expected task loss as the energy function:

$$E(\mathbf{x}, a; \theta) = \mathbb{E}_{p(y|\mathbf{x};\theta)}f(y, a), \tag{2}$$

where $p(y|\mathbf{x}; \theta)$ is the predictive distribution of an uncertainty-aware neural network [19, 31].

Eq. 2 creates an explicit energy-based differentiable layer as a surrogate to the original optimization problem. Compared with the two-stage model, it builds a direct connection between the input features $\mathbf{x}$ and decision variable $a$ and thus is more tailored to the downstream task. Compared with pure end-to-end architectures, the task-based surrogate energy function explicitly leverages the algorithmic structure of the optimization problem and thus saves a lot of learning. This energy-based surrogate function also has a intuitive interpretation. When the decision $a$ is in a region with smaller expected task loss, it has lower energy and higher probability; when it leads to higher expected task loss, it has higher energy and lower probability.

Since we have the ground truth for the parameters $y$ in the training data, the feature-decision pairs $\mathcal{D}_a = \{(\mathbf{x}_i, a_i^*)\}_{i=1}^N$ can be easily constructed by solving $a_i^* = \arg\min_{a_i \in C} f(y_i, a_i)$ for each $(\mathbf{x}_i, y_i)$ using any off-the-shelf optimization solvers. Note that the construction of such feature-action training pairs needs to be done only once during preprocessing (Fig. 3). Then, we minimize the negative log-likelihood (NLL) of all the feature-decision pairs:

$$\ell_{\text{MLE}} = -\mathbb{E}_{(\mathbf{x},a^*)\sim\mathcal{D}_a}q(a^*|\mathbf{x}; \theta) = \mathbb{E}_{(\mathbf{x},a^*)\sim\mathcal{D}_a}\left(\mathbb{E}_{p(y|\mathbf{x};\theta)}f(y, a^*) + \log(Z(\mathbf{x}; \theta))\right). \tag{3}$$

By minimizing the NLL, we are essentially minimizing the energy of the optimal actions while maximizing the energy of other points. Thus the end-to-end stochastic programming problem is translated to learning a neural network that outputs the smallest energy for the optimal actions. This new perspective avoids the need of solving and differentiating through the optimization problem at every training iteration, but meanwhile tailors the model for the downstream decision making task.

Our energy-based formulation can also be interpreted as a non-sequential maximum entropy inverse reinforcement learning (MaxEN-IRL) model [53, 52]. From the IRL perspective, the input features $\{\mathbf{x}_i\}_{i=1}^N$ can be interpreted as environment states, the feature-decision pairs $\{(\mathbf{x}_i, a_i)\}_{i=1}^N$ as expert demonstrations, and the negative expected task loss $-\mathbb{E}_{p(y|\mathbf{x};\theta)}f(y, a)$ as reward. Eq. 3 is then equivalent to maximizing the likelihood of the expert demonstrations (optimal decisions) to recover the hidden reward function.

## 3.2 Augmenting Energy-Based Objective with Distribution Regularizer

One limitation of learning with EBM-based likelihood for the optimization problem is that we model only one data point $a^*$ for each conditional distribution $q(a|\mathbf{x}; \theta)$. Minimizing the NLL for a single data point ignores the overall matchness between the landscape of original optimization problem and that of the EBM-based probability density, which can easily cause overfitting. To learn the overall

shape of the energy function better, we propose a distribution-based regularizer which augments the energy-based objective from a global training perspective.

The distribution-based regularizer is based on minimizing the distance between the model distribution and an oracle posterior distribution. Specifically, we assume that $a$ follows a prior distribution $p(a)$. With the ground-truth label $y$, the posterior distribution of $a$ is then given by $p(a|y) = \frac{p(a)\exp(-f(y,a))}{Z(y)}$ when we define the task loss as the negative log-likelihood, $i.e., -\log p(y|a) = f(y,a)$. However, such an expression depends on the ground-truth label, which cannot be obtained at test time. We let the model distribution $q(a|\mathbf{x};\theta)$ mimic this oracle posterior $p(a|y)$ distribution by minimizing their KL-divergence:

$$\ell_{\text{KL}} = \mathbb{E}_{(\mathbf{x},y)\sim\mathcal{D}}\text{KL}[p(a|y)||q(a|\mathbf{x};\theta)] = \mathbb{E}_{(\mathbf{x},y)\sim\mathcal{D}}\left(-\mathbb{E}_{p(a|y)}\log q(a|\mathbf{x};\theta) - \mathcal{H}(p(a|y))\right), \quad (4)$$

where $\mathcal{H}(\cdot)$ denotes entropy of the probability distribution. Our final training loss is a weighted combination of the NLL and the distribution based regularizer:

$$\ell_{\text{Total}} = \ell_{\text{MLE}} + \lambda\ell_{\text{KL}}, \quad (5)$$

where $\lambda$ is a hyper-parameter. In this final objective, the KL-divergence loss and the NLL loss complement each other (Fig. 2). The NLL loss is a local training strategy while the KL-divergence loss learns the energy function from a global level. Specifically, the NLL loss helps the model capture the location of the optimal decision better but ignores the overall energy shape. The distribution-based regularizer essentially adds more anchor points besides the optimal decisions and thus helps fit the overall energy shape.

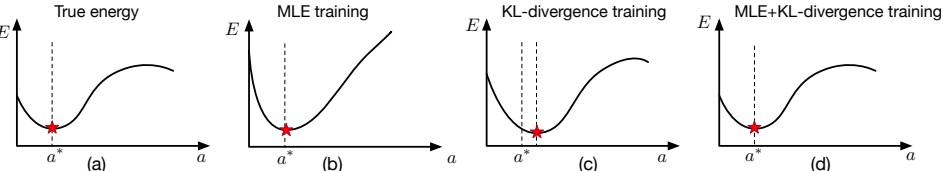

Figure 2: Different training objectives of energy-based optimization. (a) the ground-truth energy-based landscape; (b) MLE training captures the location of the optimum but ignores the overall energy shape; (c) KL-divergence training learns the overall energy shape but captures a blurry optimum location; (d) Our MLE+KL-divergence training wins the best of both worlds.

## 3.3 Training with Contrastive Divergence and Self-normalized Importance Sampling

Optimizing Eq. 5 requires evaluating the partition function and computing the KL-divergence between two continuous distributions which typically involves intractable integrals. In this subsection, we propose to use a self-normalized importance sampler based on mixture of Gaussians to estimate the gradient of the model parameters efficiently. First, we derive (see supplementary for details) the gradient of the training loss with respect to the model parameters $\theta$ as:

$$\begin{aligned}
\frac{\partial\mathcal{L}_{\text{Total}}}{\partial\theta} =& \mathbb{E}_{(\mathbf{x},a^*)\sim\mathcal{D}_a}\left(\frac{\partial E(a^*,\mathbf{x};\theta)}{\partial\theta} - \mathbb{E}_{q(\tilde{a}|\mathbf{x};\theta)}\frac{\partial E(\tilde{a},\mathbf{x};\theta)}{\partial\theta}\right) \\
&+ \lambda\mathbb{E}_{(\mathbf{x},y)\sim\mathcal{D}}\left(\mathbb{E}_{p(\hat{a}|y)}\frac{\partial E(\hat{a},\mathbf{x};\theta)}{\partial\theta} - \mathbb{E}_{q(\tilde{a}|\mathbf{x};\theta)}\frac{\partial E(\tilde{a},\mathbf{x};\theta)}{\partial\theta}\right).
\end{aligned} \quad (6)$$

As can be seen, the gradient can be estimated by sampling from the model distribution $q(a|\mathbf{x};\theta)$ and oracle distribution $p(a|y)$. Unfortunately, we cannot easily draw samples because of the unnormalized constant. Existing methods usually resort to MCMC methods (*e.g.*, Langevin dynamics) to use this gradient estimator. However, MCMC is an iterative process and can be time consuming. To improve the training efficiency, we propose to use a self-normalized importance sampler based on a Gaussian mixture proposal to estimate the gradient.

Specifically, for each $\mathbf{x}$, we first sample a set of $M$ candidates $\{a^m\}_{m=1}^M$ from a proposal distribution $\pi(a|\mathbf{x})$, and then sample $\tilde{a}$ from the empirical distribution located at each $a^m$ and weighted proportionally to $\exp(-E(a|\mathbf{x};\theta))/\pi(a|\mathbf{x})$. To reduce the variance of the self-normalized importance sampler, we propose to use a mixture of $K$ Gaussians which are centered at the location of the corresponding optimal decision as the proposal distribution: $\pi(a|\mathbf{x}) = \frac{1}{K}\sum_{i=1}^K \mathcal{N}(a^*;\sigma_k)$, where $K$ and

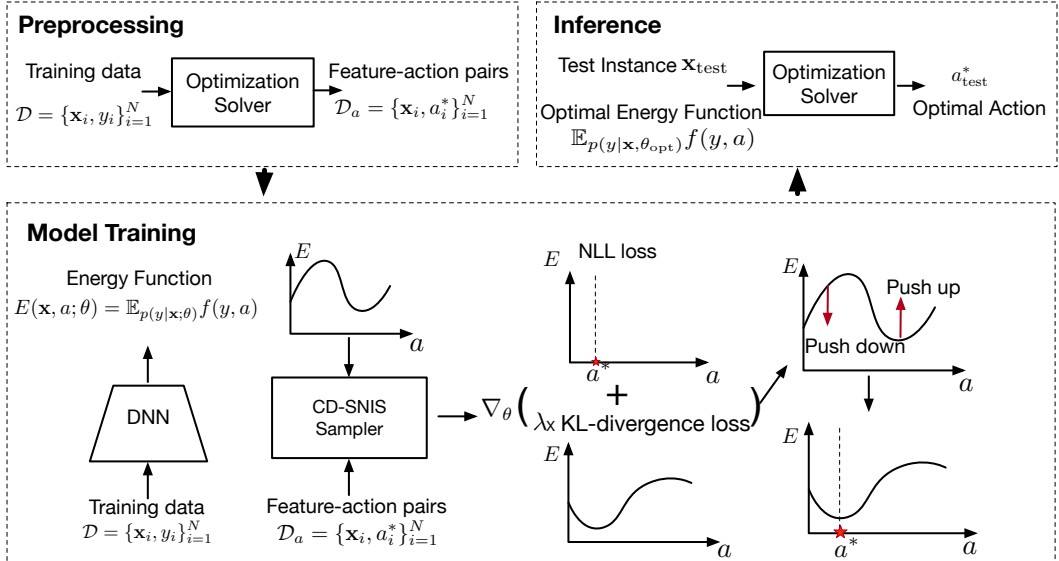

Figure 3: Illustration of SO-EBM: end-to-end stochastic optimization with energy-based model.

$\{\sigma_k\}_{k=1}^K$ are hyper-parameters. This mixture of Gaussian based self-normalized importance sampler enjoys great computational efficiency. We only need to estimate the gradient of the energy-based surrogate layer by drawing samples from a simple mixture of Gaussians, instead of solving and differentiating through the optimization problem as in DFL. This significantly reduces the training time compared with DFL as shown in Section 4. Further, locating the proposal distribution at the optimal decisions mimics the contrastive divergence method [22, 15] used in Langevin dynamics where the MCMC chain starts from the training data. This makes the proposal distribution close to the model distribution and thus has better sample efficiency. The sampling procedure from the oracle distribution $p(a|y)$ is similar with $q(a|\mathbf{x};\theta)$.

When the expected task loss has no analytical expression, we can draw multiple samples from $p(y|\mathbf{x};\theta)$ to estimate the expectation and use reparameterization trick [29, 23, 36, 42] to make it differentiable. Finally, the model can be efficiently trained via gradient-based method, such as Adam [28]. The detailed training procedure is given in Alg. 1 in the supplementary.

*Model inference.* With the optimal model parameter $\theta_{\mathrm{opt}}$, we draw samples from $q(a|\mathbf{x}_{\mathrm{test}};\theta_{\mathrm{opt}})$ for an unseen test instance $\mathbf{x}_{\mathrm{test}}$. However, in the real-world applications, we usually only need the most optimal decision. This can be obtained by solving $\arg\min_{a_{\mathrm{test}}^* \in C} \mathbb{E}_{p(y|\mathbf{x}_{\mathrm{test}};\theta_{\mathrm{opt}})} f(y, a)$ with any existing black-box optimization solver such as CVXPY [12, 2] and Pyomo [5, 21] (Fig. 3).

## 4 Experiments

In this section, we empirically evaluate SO-EBM. We conduct experiments in three applications: (1) Load forecasting and generator scheduling where the expected task loss has a closed-form expression; (2) Resource allocation for COVID-19 where the expected task loss has no closed-form expression; (3) Adversarial behavior learning in network security with a non-convex optimization objective. Finally, we do ablation studies to show the effect of each component in SO-EBM.

### 4.1 Load Forecasting and Generator Scheduling

In this task, a power system operator needs to decide how much electricity $a \in \mathbb{R}^{24}$ to schedule for each hour in the next 24 hours to meet the actual electricity demands. The optimization objective is a combination of an under-generation penalty, an over-generation penalty, and a mean squared loss between supplies and demands:

$$\text{minimize}_{a \in \mathbb{R}^{24}} \sum_{i=1}^{24} \mathbf{E}_{y \sim p(y|x;\theta)} [\gamma_s [y_i - a_i]_+ + \gamma_e [a_i - y_i]_+ + \frac{1}{2}(a_i - y_i)^2] \tag{7}$$
$$\text{subject to} \quad |a_i - a_{i-1}| \le c_r \quad \forall i,$$

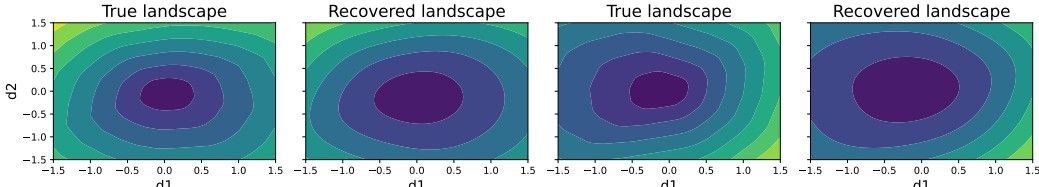

Figure 4: Ground-truth and SO-EBM recovered landscapes of the energy function in the power generator scheduling task. Darker colors represent lower energy in the heat maps. For a test sample $\mathbf{x}_{\text{test}}$, we choose the corresponding optimal action $a^*_{\text{test}}$ as the center point and select two random direction vectors $v_1$ and $v_2$ to plot the energy landscape, $i.e., E'(d_1, d_2) = E(\mathbf{x}, a^* + d_1 v_1 + d_2 v_2)$.

where $[\cdot]_+ = \max\{v, 0\}$. Usually, the penalty coefficients satisfy $\gamma_e \gg \gamma_s$ since under-generation is more serious than over-generation. The quadratic regularization term indicates the preference for generation schedules that closely match actual demands. The ramping constraint $c_r$ restricts the change in generation between consecutive timepoints, which reflects physical limitations associated with quick changes in electricity output levels.

**Experiment Setup**. We forecast the electricity demands $y \in \mathbb{R}^{24}$ over all 24 hours of the next day using a 2-hidden layer neural network. We assume $y_i$ is a Gaussian with mean $\mu_i$ and variance $\sigma_i^2$; as such, the expectation in the optimization objective can be computed analytically. The input features $\mathbf{x}$ is a 150-dimensional vector including the past day's electrical load and temperature, the next day's temperature forecast, non-linear functions of the temperatures, binary indicators of weekends or holidays, and yearly sinusoidal features. Following [13], we set $\gamma_s = 0.4$, $\gamma_e = 50$ and $c = 0.4$.

We compare with the following baselines on this task: (1) A two-stage predict-then-optimize model trained with negative likelihood loss for the prediction task. (2) Decision-focused learning with the QPTH solver [13]. It uses sequential quadratic programming (SQP) to iteratively approximate the resultant convex objective as a quadratic objective, iterates until convergence, and then computes the necessary Jacobians using the quadratic approximation at the solution. (3) DFL with cvxpylayers [1] (DFL-CVX) which provides a differentiable layer with disciplined parameterized programming (DPP) grammar. Since the analytical expectation of Eq. 7 cannot be written in DPP, we use Monte Carlo sampling to estimate the expectation for this baseline. (4) Policy-net. It direct maps from the input features to the decision variables by minimizing the task loss using supervised learning [13]. Our supplementary provides more details of the model parameters.

**Results.** Fig. 5 shows the end task loss for all the methods. Our method SO-EBM outperforms all the baselines with a significant reduction of training time. Compared with the strongest baseline DFL-QPTH, SO-EBM improves the task loss by 7.3%. The improvement is because DFL-QPTH needs to use SQP to iteratively obtain the solutions for non-quadratic optimization problems. Differentiating through all the steps of SQP is prohibitively expensive in memory and time. To address this issue, existing works [13] differentiate through just the last step of SQP to obtain approximate gradients. The inaccurate gradients

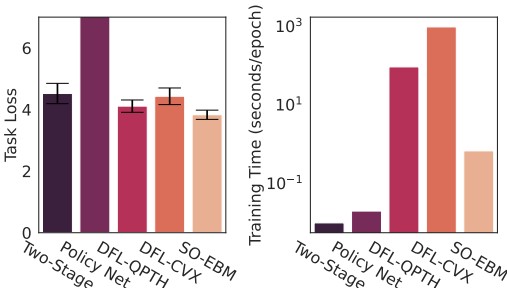

Figure 5: Results on power generator scheduling.

may accrue during training and thus hurt decision quality. This shows that our explicit differentiable energy-based function is an effective surrogate for the original implicit optimization layer. In term of efficiency, SO-EBM is more than **136** times faster than DFL-QPTH (0.68 second/epoch *v.s.* 93.12 second/epoch) in training. This is because we only need to draw samples from a simple mixture of Gaussians to estimate the model parameters instead of solving and differentiating through the optimization problem at every training iteration. DFL-CVX is even slower than DFL-QPTH since it needs to use sample average estimation to draw multiple samples from $p(y|\mathbf{x}; \theta)$, which results in a much more complicated optimization objective. This is also likely the reason why DFL-CVX under-performs DFL-QPTH in terms of task loss here. For Policy-Net, we have tuned its hyper-parameters extensively but still cannot achieve good performance on this task (2x-3x larger task loss than the two-stage baseline). This is not surprising: as aforementioned, Policy-Net is a pure end-to-end model

that needs a large amount of data to rediscover the algorithmic structure of the optimization task. In contrast, DFL and So-EBM model the predict-and-optimize structure in the model design, which save learning and are more data-efficient.

Fig. 4 shows the ground-truth and So-EBM learned landscapes of the energy function. As we can see, So-EBM can recover the landscape of the original optimization objective effectively though with a small discrepancy. The small discrepancy is expected since the ground-truth landscape is computed by directly using the ground-truth label, while So-EBM uses the uncertainty-aware neural network to first forecast the distribution of the label and then uses the predictive distribution to compute the expected task loss as the energy function.

## 4.2 Resource Allocation for COVID-19

As seen in the COVID-19 pandemic, an increasing number of infected patients leads to increasing demand for medical resources; which makes it challenging for policymakers and epidemiologists to plan ahead. Mechanistic epidemiological models based on Ordinary Differential Equations (ODEs) are often used to capture and forecast the dynamics of epidemics including for the COVID-19 pandemic [35, 24, 20, 10]. Such forecasts are typically then used as guidance to help plan for future resource allocation [38, 18, 43, 25]. In this task, we study the optimization problem of hospital bed preparation for COVID-19, one of the most common and important tasks epidemiologists focus on during the pandemic [3, 7]. We need to decide how many beds $a \in \mathbb{R}^7$ are needed to prepare for the next week based on the forecasted number of hospitalized patients $y \in \mathbb{R}^7$. The optimization objective is a combination of linear and quadratic costs, which accounts for over-preparations $[a - y]_+$ and under-preparations $[y - a]_+$ in the next 7 days over ODE-derived dynamics:

$$\text{minimize}_{a \in \mathbb{R}^7} \sum_{i=1}^{7} c_b [y_i - a_i]_+ + c_h [a_i - y_i]_+ + q_b ([y_i - a_i])_+^2 + q_h ([a_i - y_i])_+^2. \quad (8)$$

**Experiment Setup**. We use the SEIR+HD ODE model proposed in [24] to capture the dynamics of the COVID-19 pandemic, which has been used in policy studies for interventions. The model is driven by a key parameter: the transmission rate $\beta$ that reflects the probability of disease transmission. The forecasting model is a two-layer gated recurrent unit (GRU) [6] which takes the number of people in each state of the SEIR+HD model in the last 21 days as input features and outputs the transmission rate $\beta$ for the next 7 days. We assume that $\beta$ follows a Gaussian distribution. Since SEIR-HD is a complex non-linear ODE system, there is no closed expression for the distribution of the number of hospitalized patients $y$. Hence, we choose to sample from the forecasted distribution of $\beta$ 100 times and then simulate the SEIR-HD model to obtain the empirical distribution of the number of hospitalized patients $y$.

**Results**. Fig. 6 shows the results on the resource allocation for COVID-19 task. Both So-EBM and DFL significantly outperforms the two-stage model in terms of the task loss, yielding 10.9%+ improvement. DFL-QPTH is not included here because it needs to use SQP to iteratively approximate the objective and is much slower than DFL-CVX in this task. The performance of So-EBM and DFL are similar, however, the training time of So-EBM is 4.3 times faster than DFL. The time improvement of So-EBM is not as large as on the energy scheduling task because all the methods need to compute the complex ODE equations in the forward pass which is a time consuming part. Therefore, considering the overall training efficiency, we argue that So-EBM is a better option than DFL on this task in practice.

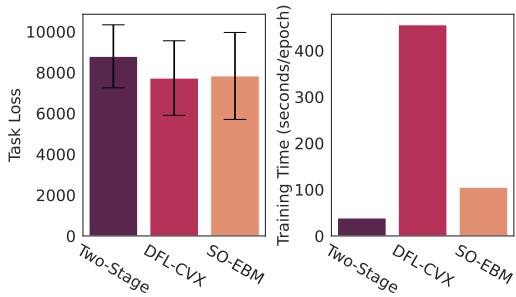

Figure 6: Results on Covid-19 resource allocation.

## 4.3 Network Security Game

In this task, we study stochastic optimization in network security games [49]. Given a network $G = (V, E)$, a source node $s \in V$, and a set of target nodes $T \in V$, a *network security game* (NSG) [50, 17, 44] defines the min-max game where the attacker attempts to travel from $s$ to any $t \in T$ while the defender places checkpoints on certain number of edges in the graph. Each of the

target node has a reward $u(t)$ should the attacker reach them. The defender first chooses a mixed strategy. Having observed the defender's mixed strategy but not the sampled pure strategy, the attacker attempts to choose a path. To minimize the attacker's scores, the defender can try to predict the attacker's path decisions with node features and their past decisions, since the attacker is not perfectly rational in reality. Suppose that $\mathbf{a}$ defines the placement of checkpoints by the defender where $\mathbf{a}_e$ is the probability that edge $e$ is covered. The attackers will perform a random walk on the graph, generating a path $r$, stopping at either a target or a checkpoint. The transition between two nodes is determined by the defender's strategy $\mathbf{a}$ and node-level parameters $\mathbf{y}$, where $\mathbf{y}_u$ defines the "attractiveness" of node $u$, representing the attacker's idiosyncratic preferences. The defender wants to choose $\mathbf{a}$ to maximize its expected utility: $\max_{\mathbf{a}} \mathbb{E}_{T \sim \mathbf{a}} \mathbb{E}_{r \sim p(r|\mathbf{a},\mathbf{y})} - g(r)$, where $T$ is covered edges sampled according to $\mathbf{a}$, $g(r) = u(t)$ if the attacker reaches target $t$ with path $r$ and 0 if the attacker is stopped at a checkpoint. The defender's objective comes with the constraint $\sum \mathbf{a} \leq k$, where $k = 3$ represents the resources available to the defender. The optimization objective is non-convex due to the irrational strategy of the attacker [48].

**Setup.** We follow the setup of [49], see supplementary for details.

**Results**. Fig. 7 shows the results on the adversarial network security game task. Since the original DFL fails on this large scale problem, we compare a block-variable sampling approach specialized to this task (DFL-Block)[48]. Surrogate [49] performs a linear dimension reduction for DFL to speed up the training time and improve the performance by smoothing the training landscape. We did not present the results of Surrogate in the previous two tasks because the numbers of decision variables there are relatively small and it has even worse performance than the original DFL approach. As we can see,

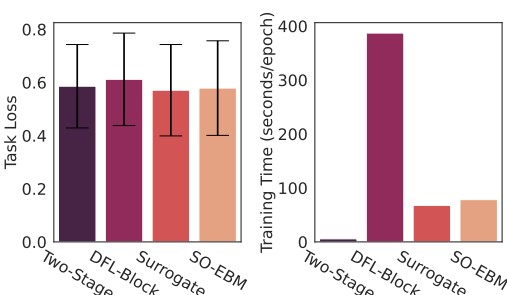

Figure 7: Results on network security game.

SO-EBM can achieve competitive task loss with Surrogate and outperforms DFL-block and the two-stage method. The increase of training time in our method is mainly because every evaluation of the defender's utility requires a matrix inverse; our method needs to evaluate the utility for all the samples drawn from the proposal distribution which results a larger matrix to inverse. However, this issue can be mitigated by using some advanced matrix inverse algorithm [9, 32]. DFL-Block is even worse than the two-stage model because it back-propagates through randomly sampled variables which results in inaccrurate gradient estimation.

## 4.4 Ablation Study

We investigate the effectiveness of the coupled training objective and alternative EBM training algorithms via ablation studies on the load forecasting and generator scheduling task. Table 1 shows the results. Our findings can be summarized as follows: (1) Both the MLE and KLD training objectives work and outperform the two-stage model. This verifies the effectiveness of using the energy-based surrogate function to approximate the optimization landscape. (2) By

| Method | Task loss |
|---|---|
| Two-stage | $4.52 \pm 0.33$ |
| SO-EBM w/o MLE | $4.31 \pm 0.17$ |
| SO-EBM w/o KLD | $3.89 \pm 0.18$ |
| SO-EBM w/ CD-LD | $3.85 \pm 0.13$ |
| **SO-EBM** | $\mathbf{3.83 \pm 0.15}$ |

Table 1: Ablation study on power generator scheduling.

dropping either the MLE or KLD term, we observe a performance degradation in our method. Specifically, without the MLE term, the task loss increases by $13\%$; without KLD term, the task loss increases by $2\%$. The larger degradation when simply minimizing the KLD term is possibly because accurately recover the entire optimization landscape is too difficult and it needs the MLE term to help capture the location of the optimal decision. (3) Training SO-EBM with Contrastive Divergence based Langevin Dynamics (CD-LD) [15] achieves similar performance but incurs longer training time (1.47 second/epoch *v.s.* 0.68 second/epoch). This phenomenon is likely because the optimization problem is relatively low-dimensional, thus importance sampling works well and enjoys better efficiency in this regime. However, even with CD-LD, SO-EBM is still 63 times faster than DFL-QPTH (1.47 second/epoch *v.s.* 93.12 second/epoch).

We also investigate the impact of training data size for each method. Fig. 8 provides the task losses and training time under different ratios of training data on the load forecasting and generator scheduling task. As we can see, our method outperforms the baselines constantly except for the ratio of $0.05$. When the ratio is below $0.05$, all the methods cannot work properly due to the extremely low resource. The superior performance of our method is because we design the energy function as the expected task loss, which leverages the algorithmic structure inherent in the optimization problem. Hence, our method is not very data demanding. In terms of efficiency, our method reduces the training time

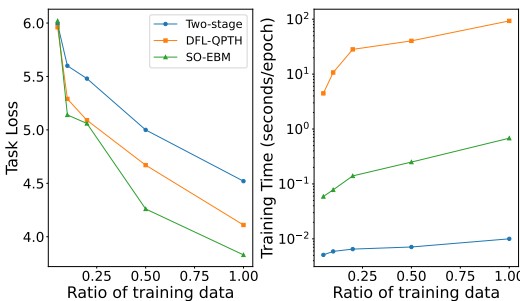

Figure 8: Task losses and training time under different ratios of training data on power generator scheduling.

significantly compared with DFL-QPTH across different amounts of training data.

## 5 Additional Related Work

The DFL works described in Section 2 focus on continuous optimization. There are also studies that extend DFL for combinatorial optimization. [51] relaxes the discrete decision into its continuous counterpart and adds a quadratic regularization term to avoid vanishing gradients. [37] proposes a log barrier regularizer and differentiates through the homogeneous self-dual embedding. [39] proposes a noise contrastive objective by maximizing the distance between the optimal solution and noisy samples. Our framework may be also extended for combinatorial problems by using discrete energy-based model [11]. The SPO+ loss [16] has been proposed to measure the prediction errors against optimization objectives, but it is only applicable to linear programming. ProjectNet [8] approximately solves the linear programming problems using a differentiable projection architecture. Our work is also related to learning to optimize [47, 34, 26], which learns a policy network that solves optimization problems using supervised or reinforcement learning. However, these pure approaches need to rediscover the structure of the optimization problem and thus data-inefficient.

## 6 Limitations and Discussion

We focused on addressing the scalability and generality of existing decision-focused learning (DFL) for end-to-end stochastic optimization. We argue that these deficiencies stem from the reliance of implicitly differentiable optimization layers based on KKT conditions. As a remedy, we circumvent such deficiencies by replacing the implicit optimization layer with a newly parameterized energy-based surrogate function. We proposed a coupled training objective to encourage the energy-based surrogate well approximate the optimization landscape, as well as an efficient training procedure based on self-normalized importance sampling. Empirically, we demonstrated that our energy-based model is effective in a wide range of stochastic optimization problems with either convex or nonconvex objectives. It can achieve better or comparable performance than state-of-the-art DFL methods for stochastic optimization, while being several times or even orders of magnitude faster.

We discuss limitations and possible extensions of SO-EBM: (1) *Handling more complex constraints*. Our method handles the constraints implicitly through the pre-processing step and explicitly through the inference step. However, when the feasible space is extremely small, training the constrained EBM may have more benefits. To train EBMs in the constrained space, one direction is to project samples into the feasibility space. Another direction is to explore adding soft constraints to the energy function during training, *e.g.*, Augmented Lagrangian penalty and Barrier penalty. (2) *More effective training methods*. Our method is a general framework for end-to-end stochastic programming problem based on EBM. There are a number of training techniques [45] that can be plugged into our framework and we can further improve the task performance using the advanced EBM training algorithms [14, 40].

**Acknowledgments:** We thank the anonymous reviewers for their helpful comments. This work was supported in part by the NSF (Expeditions CCF-1918770, CAREER IIS-2028586, IIS-2027862, IIS-1955883, IIS-2106961, IIS-2008334, CAREER IIS-2144338, PIPP CCF-2200269), CDC MInD program, faculty research award from Facebook and funds/computing resources from Georgia Tech.

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
