# Supplementary Material for End-to-End Stochastic Optimization with Energy-Based Model

**Lingkai Kong**   **Jiaming Cui**   **Yuchen Zhuang**   **Rui Feng**
**B. Aditya Prakash**   **Chao Zhang**
College of Computing
Georgia Institute of Technology
{lkkong,jiamingcui1997,yczhuang,rfeng,badityap,chaozhang}@gatech.edu

## 1   Energy-based Model

The underpinning of energy-based models (EBMs) [7] is the fact that any probability density $p(\mathbf{x}; \theta)$ can be expressed as:

$$p(\mathbf{x}; \theta) = \frac{\exp(-E(\mathbf{x}; \theta))}{Z(\theta)},$$

where $E(\mathbf{x}; \theta) : \mathbb{R}^D \to \mathbb{R}$ is a nonlinear regression function that maps each input data to an energy scalar; and $Z(\theta) = \int \exp(-E(\mathbf{x}; \theta)) d\mathbf{x}$ is a normalizing constant, also known as partition function. In this energy-based parameterization, data points with high likelihood have low energy, while data points with low likelihood have high energy. Due to its simplicity and flexibility, EBMs have been widely used in many learning tasks, including image generation [2, 13], out-of-distribution detection [3, 8], and density estimation [9, 12].

## 2   Derivation for Eq. 6

To obtain the gradient of the model parameters, we first derive $\frac{\partial \log(Z(\mathbf{x}; \theta))}{\partial \theta}$:

$$
\begin{aligned}
\frac{\partial \log Z(\mathbf{x}; \theta)}{\partial \theta} &= \frac{\partial \log \int \exp(-E(a, \mathbf{x}; \theta)) da}{\partial \theta} \\
&= \left( \int \exp(-E(a, \mathbf{x}, a; \theta)) da \right)^{-1} \frac{\partial \int \exp(-E(\mathbf{x}, a; \theta)) da}{\partial \theta} \\
&= \left( \int \exp(-E(a, \mathbf{x}, a; \theta)) da \right)^{-1} \frac{\int \partial \exp(-E(\mathbf{x}, a; \theta)) da}{\partial \theta} \\
&= \left( \int \exp(-E(a, \mathbf{x}, a; \theta)) da \right)^{-1} \int \exp(-E(a, \mathbf{x}; \theta)) \left( -\frac{\partial E(a, \mathbf{x}; \theta)}{\partial \theta} \right) da \\
&= \int \left( \int \exp(-E(a, \mathbf{x}, a; \theta)) da \right)^{-1} \exp(-E(a, \mathbf{x}; \theta)) \left( -\frac{\partial E(a, \mathbf{x}; \theta)}{\partial \theta} \right) da \\
&= \int \frac{\exp(-E(a, \mathbf{x}; \theta))}{Z(\mathbf{x}; \theta)} \left( -\frac{\partial E(a, \mathbf{x}; \theta)}{\partial \theta} \right) da \\
&= \int q(a | \mathbf{x}; \theta) \left( -\frac{\partial E(a, \mathbf{x}; \theta)}{\partial \theta} \right) da \\
&= -\mathbb{E}_{q(a | \mathbf{x}; \theta)} \frac{\partial E(a, \mathbf{x}; \theta)}{\partial \theta}
\end{aligned}
$$

36th Conference on Neural Information Processing Systems (NeurIPS 2022).

Then the gradient of the model parameters is given by:

$$
\begin{aligned}
\frac{\partial \mathcal{L}_{\text{Total}}}{\partial \theta} &= \frac{\partial \mathcal{L}_{\text{MLE}}}{\partial \theta} + \lambda \frac{\partial \mathcal{L}_{\text{KL}}}{\partial \theta} \\
&= \mathbb{E}_{(\mathbf{x},a^*) \sim \mathcal{D}_a} \left( \frac{\partial E(a^*, \mathbf{x}; \theta)}{\partial \theta} + \frac{\partial \log(Z(\mathbf{x}; \theta))}{\partial \theta} \right) \\
&\quad + \lambda \mathbb{E}_{(\mathbf{x},y) \sim \mathcal{D}} \left( -\frac{\partial \mathbb{E}_{p(\hat{a}|y)} \log q(\hat{a}|\mathbf{x}; \theta)}{\partial \theta} - \underbrace{\frac{\partial \mathcal{H}(p(a|y))}{\partial \theta}}_{0} \right) \\
&= \mathbb{E}_{(\mathbf{x},a^*) \sim \mathcal{D}_a} \left( \frac{\partial E(a^*, \mathbf{x}; \theta)}{\partial \theta} + \frac{\partial \log(Z(\mathbf{x}; \theta))}{\partial \theta} \right) \\
&\quad + \lambda \mathbb{E}_{(\mathbf{x},y) \sim \mathcal{D}} \left( \mathbb{E}_{p(\hat{a}|y)} \frac{\partial E(\hat{a}, \mathbf{x}; \theta)}{\partial \theta} + \frac{\partial \mathbb{E}_{p(\hat{a}|y)} \log(Z(\mathbf{x}; \theta))}{\partial \theta} \right) \\
&= \mathbb{E}_{(\mathbf{x},a^*) \sim \mathcal{D}_a} \left( \frac{\partial E(a^*, \mathbf{x}; \theta)}{\partial \theta} + \frac{\partial \log(Z(\mathbf{x}; \theta))}{\partial \theta} \right) \\
&\quad + \lambda \mathbb{E}_{(\mathbf{x},y) \sim \mathcal{D}} \left( \mathbb{E}_{p(\hat{a}|y)} \frac{\partial E(\hat{a}, \mathbf{x}; \theta)}{\partial \theta} + \frac{\partial \log(Z(\mathbf{x}; \theta))}{\partial \theta} \right) \\
&= \mathbb{E}_{(\mathbf{x},a^*) \sim \mathcal{D}_a} \left( \frac{\partial E(a^*, \mathbf{x}; \theta)}{\partial \theta} - \mathbb{E}_{q(\tilde{a}|\mathbf{x};\theta)} \frac{\partial E(\tilde{a}, \mathbf{x}; \theta)}{\partial \theta} \right) \\
&\quad + \lambda \mathbb{E}_{(\mathbf{x},y) \sim \mathcal{D}} \left( \mathbb{E}_{p(\hat{a}|y)} \frac{\partial E(\hat{a}, \mathbf{x}; \theta)}{\partial \theta} - \mathbb{E}_{q(\tilde{a}|\mathbf{x};\theta)} \frac{\partial E(\tilde{a}, \mathbf{x}; \theta)}{\partial \theta} \right).
\end{aligned}
$$

## 3  Training Algorithm for SO-EBM

We adopt gradient-based method such as Adam [5] to update the model parameters. At each epoch, we need to draw samples from the distributions $q(a|\mathbf{x}; \theta)$ and $p(a|y)$ to estimate the gradient of the model parameters. Based on the self-normalized importance sampler, we first sample a set of $M$ candidates $\{a^m\}_{m=1}^M$ from a proposal distribution $\pi(a|\mathbf{x})$, and then sample from the empirical distribution located at each $a^m$ and weighted proportionally to $\exp(-E(a|\mathbf{x};\theta))/\pi(a|\mathbf{x})$ and $\exp(-f(y,a))/\pi(a|\mathbf{x})$. Then, we compute the gradient of the model parameters using these empirical samples according to Eq. 6. We summarize the overall training procedure in Alg. 1.

## 4  Experimental Details

### 4.1  Load Forecasting and Generator Scheduling

**Closed-form expression of the expected objective.** Given the electricity demand $y \in \mathbb{R}^{24}$ for the next 24 hours, the electricity scheduling problem is to decide how much electricity $a \in \mathbb{R}^{24}$ to schedule. The optimization objective is given by

$$
\begin{aligned}
\text{minimize}_{a \in \mathbb{R}^{24}} \sum_{i=1}^{24} \mathbf{E}_{y \sim p(y|x;\theta)} [ \gamma_s [y_i - a_i]_+ + \gamma_e [a_i - y_i]_+ + \frac{1}{2}(a_i - y_i)^2 ] \\
\text{subject to} \quad |a_i - a_{i-1}| \le c_r \quad \forall i,
\end{aligned}
$$

where $c_r$ is the ramp constraint and $\gamma_s$, $\gamma_e$ are the under-generation penalty and over-generation penalty respectively.

---

**Algorithm 1** Training of SO-EBM

---

**Require:** Training data $\mathcal{D} = \{(\mathbf{x}_i, y_i)\}_{i=1}^N$, task loss function $f(y, a)$, model $\mathbf{M}$ parameterized with parameter $\theta$

1: Construct feature-decision pairs $\mathcal{D}_a = \{(\mathbf{x}_i, a_i^*)\}$ by solving $a^* = \arg\min_{a \in C} f(y_i, a)$ for each $(\mathbf{x}_i, y_i)$ in $\mathcal{D}$

2: $E(\mathbf{x}, a; \theta) \triangleq -\mathbb{E}_{p(y|\mathbf{x};\theta)} f(y, a)$

3: **for** # training iterations **do**

4:     Sample a mini-batch $\mathcal{B}_y$ from $\mathcal{D}$ and $\mathcal{B}_a$ from $\mathcal{D}_a$

5:     **for** For each $(\mathbf{x}_i, a_i^*)$ and $(\mathbf{x}_i, y_i)$ in $\mathcal{B}_a$ and $\mathcal{B}_y$ **do**   ▷ Self-normalized importance Sampler

6:         $\pi(a; \mathbf{x}_i) \triangleq \frac{1}{K} \sum_{k=1}^K \mathcal{N}(a_i^*; \sigma_k)$

7:         **for** $m = 1, \cdots, M$ **do**

8:             Sample $a_i^m \sim \pi(a; \mathbf{x}_i)$

9:             Compute $\tilde{w}(a_i^m) = \exp(-E(\mathbf{x}_i, a_i^m; \theta))/\pi(a_i^m; \mathbf{x}_i)$

10:            Compute $\hat{w}(a_i^m) = \exp(-f(y_i, a_i^m))/\pi(a_i^m; \mathbf{x}_i)$

11:         **end for**

12:         Compute $\tilde{Z} = \sum_{m=1}^M \tilde{w}(a_i^m)$, $\hat{Z} = \sum_{m=1}^M \hat{w}(a_i^m)$

13:         $q(\tilde{a}|\mathbf{x}_i; \theta) \triangleq \sum_{m=1}^M \frac{\tilde{w}(a_i^m)}{\tilde{Z}} \delta_{a_i^m}(\tilde{a})$

14:         $p(\hat{a}|y) \triangleq \sum_{m=1}^M \frac{\hat{w}(a_i^m)}{\hat{Z}} \delta_{a_i^m}(\hat{a})$

15:     **end for**

16:     Compute $\frac{\partial \mathcal{L}_{\text{Total}}}{\partial \theta}$ via Eq. 6 using $\mathcal{B}_y$, $\mathcal{B}_a$, $q(\tilde{a}|\mathbf{x}_i; \theta)$ and $p(\hat{a}|y)$.

17:     Update $\theta$ using ADAM

18: **end for**

---

Following [1], we assume $y_i$ follows a Gaussian distribution with mean $\mu_i$ and variance $\sigma_i$ and obtain the closed form expression:

$$\sum_{i=1}^{24} \mathbf{E}_{y \sim p(y|x;\theta)} [\gamma_s [y_i - a_i]_+ + \gamma_e [a_i - y_i]_+ + \frac{1}{2}(a_i - y_i)^2]$$

$$= \sum_{i=1}^{24} (\gamma_s + \gamma_e)(\sigma_i^2 p(a_i; \mu_i, \sigma_i^2) + (a_i - \mu_i) F(a_i, \mu_i, \sigma_i^2)) - \gamma_s(a_i - \mu_i) + \frac{1}{2}((a_i - \mu_i)^2 + \sigma_i^2).$$

(1)

where $p(a; \mu, \sigma^2)$ and $F(a; \mu, \sigma^2)$ denote the Gaussian probability density function (PDF) and cumulative distribution function (CDF), respectively with the given mean and variance. Eq. 1 is a convex function of $a$ since the expectation of a convex function is still convex.

**Model hyperparameters**. We use a two-hidden-layer neural network, where each "layer" is a combination of linear, batch norm [4], ReLU, and dropout ($p = 0.2$) layers with dimension 200. SO-EBM draws 512 samples from the proposal distribution to estimate the gradient of the model parameters. The proposal distribution is a mixture of Gaussians with 3 components where the variances are $\{0.02, 0.05, 0.1\}$. The weight parameter $\lambda$ that balances the MLE and KLD terms is set to 1. DFL-CVX draws 50 samples from $p(y|\mathbf{x}; \theta)$ to estimate the expectation of the task loss since it cannot use the closed-form expression.

**Model optimization.** We use the Adam [5] algorithm for model optimization. The number of training epochs is 100. The learning rate is selected from $\{10^{-3}, 10^{-4}, 5 \times 10^{-5}, 10^{-5}\}$ based on the task loss on the validation set. Specifically, the learning rate for the two-stage model and PolicyNet is $10^{-3}$. The learning rate for DFL-QPTH and DFL-CVX is $10^{-4}$. The learning rate for SO-EBM is $5 \times 10^{-5}$. DFL-QPTH, DFL-CVX and SO-EBM use the two-stage model as the pre-trained model for faster training convergence.

## 4.2 Resource Allocation for COVID-19

The dataset[1] contains the number of hospitalized patients in California from 3/29/2020 to 12/18/2021. We use the first 70% data as the training and validation sets, the remaining as the testing set. In the

---

[1] The dataset is available at `https://gis.cdc.gov/grasp/covidnet/covid19_5.html`

first 70% data, we randomly select 80% as the training set and the remaining is used as the validation set. For the optimization objective, we set $c_b = 10$, $c_h = 1$, $q_b = 2$ and $q_h = 0.5$.

**Model hyperparameters**. We use a two-layer gated recurrent unit (GRU) with hidden-size 128 as the forecasting model. SO-EBM draws 512 samples from the proposal distribution to estimate the gradient of the model parameters. The proposal distribution is a mixture of Gaussians with 3 components where the variances are $\{5, 10, 20\}$. The weight parameter $\lambda$ that balances the MLE and KLD terms is set to 1.

**Model optimization**. We use the Adam [5] algorithm for model optimization. The number of training epochs is 50. The learning rate is selected from $\{10^{-2}, 10^{-3}, 10^{-4}\}$ based on the task loss on the validation set. Specifically, the learning rate for the two-stage model and DFL-QPTH is $10^{-2}$. The learning rate for SO-EBM is $10^{-3}$. DFL-CVX and SO-EBM use the two-stage model as the pre-trained model for faster training convergence.

### 4.3 Network Security Game

**Dataset generaton**. Following [11, 10], we generate random geometric graphs with 100 nodes. The generated graphs have a radius of 0.2 in unit square. 5 nodes are chosen randomly as targets with rewards $u(t) \sim \mathcal{U}(5, 10)$. 5 other nodes are sampled as candidate source nodes, from which the attacker chooses uniformly. For any node not $v$ in the node set, its attractiveness score $y_v$ is proportional to its distance to the closest target in with an additional uniformly distributed noise $\mathcal{U}(-1, 1)$, representing the attacker's distinct preferences. The node features $\mathbf{x}$ are computed with a Graph convolutional network (GCN) [6]: $\mathbf{x} = \text{GCN}(y) + 0.2\mathcal{N}(0, 1)$. In this experiment, we use a randomly initialized GCN which has 4 convolutional layers and 3 fully connected layers. In this way we generate random node features correlated with their attractiveness scores and nearby node features. This is a realistic setting since neighboring locations are supposed to have similar features. The defender uses a different GCN model for prediction, which has 2 convolutional and 2 fully connected layers, ensuring that the defender's model is less complex than the true generative mechanics. With the described procedure we sample 35 random $(\mathbf{x}, y)$ pairs for the training set, 5 for the validation set, and 10 for the testing set.

**Model hyperparameters.** We follow [11] to configure the forecasting model. Specifically, we employ a two-layer GCN with output dimensions equal to 16 and average all node representations as the whole graph representation. The whole graph representation is further fed into a linear network with two fully-connected layers in which the hidden dimension is set to 32. SO-EBM draws 512 samples from the proposal distribution to estimate the gradient of the model parameters. The proposal distribution is a mixture of Gaussians with 3 components where the variances are $\{0.01, 0.05, 0.1\}$. The weight parameter $\lambda$ that balances the MLE and KLD terms is set to 1. The reparameterization size of Surrogate is set to 10 as suggested in [11].

**Model optimization.** We use the Adam [5] algorithm for model optimization. All the methods are trained for at most 100 epochs and early stopped when 3 consecutive non-improving epochs occur on the validation set. Following [11], the learning rate for the two-stage model, CVX-block and Surrogate is set to $10^{-2}$. The learning rate for SO-EBM is $10^{-3}$.

## 5 Visualization using Synthetic Dataset

To visualize whether SO-EBM can recover the landscape of the optimization objective effectively, we conduct experiments on a 2-D synthetic dataset. We generate 500 features $\mathbf{x} \in \mathbb{R}^2$ from the uniform distribution $\mathcal{U}[-2, 2]$. The label $y \in \mathbb{R}^2$ is generated by $y_i = x_i^2 + 0.03\mathcal{N}(0, 1)$ for each dimension independently. The optimization objective is:

$$\text{minimize}_{a \in \mathbb{R}^2} \mathbb{E}_{p(y|\mathbf{x};\theta)} \sum_{i=1}^{2} 3|a_i - y_i| + 0.5(a_i - 1.5)^2. \tag{2}$$

We randomly select 80% data as the training set and the remaining is used for testing. The network architecture and model hyperparameters are the same as those we used in the load forecasting and generator scheduling task.

Fig. 1 shows the ground-truth and SO-EBM learned landscapes of the optimization objectives. As we can see, SO-EBM can recover the landscape of the original optimization objective effectively though

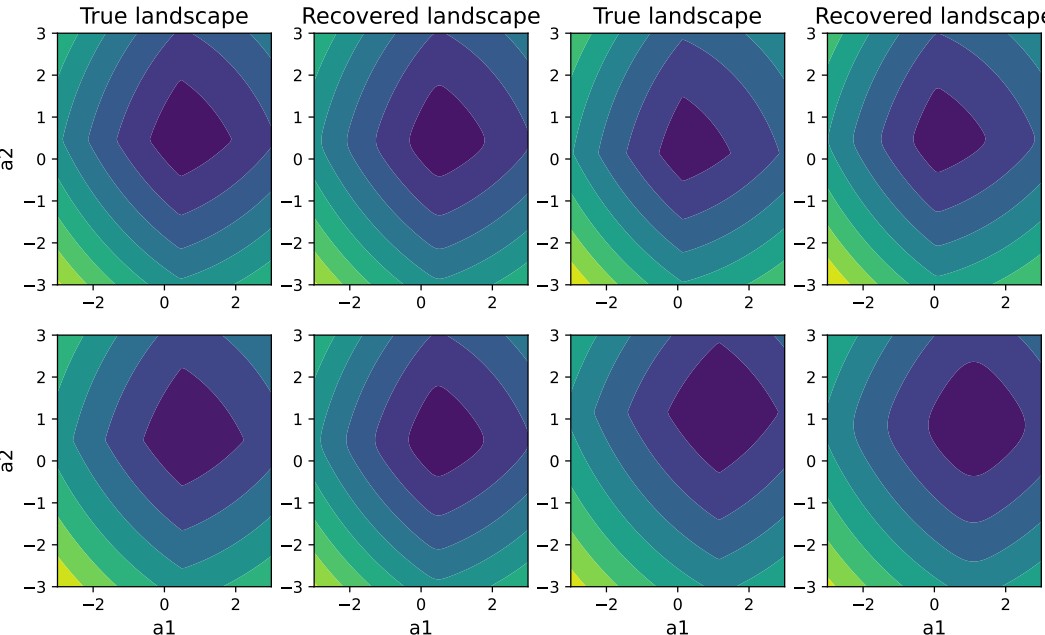

Figure 1: Ground-truth and SO-EBM recovered landscapes of the optimization objective. Darker colors represent smaller task loss in the heat maps.

with a small discrepancy. The small discrepancy is reasonable since the ground-truth landscape is computed by directly using the ground-truth label, while SO-EBM uses the uncertainty-aware neural network to first forecast the distribution of the label and then uses the predictive distribution to compute the expected task loss.

# 6 Computing Infrastructure

System: Ubuntu 18.04.6 LTS; Python 3.9; Pytorch 1.11. CPU: Intel(R) Xeon(R) Silver 4214 CPU @ 2.20GHz. GPU: GeForce GTX 2080 Ti.