# OpenReview forum: "End-to-end Stochastic Optimization with Energy-based Model"
_NeurIPS.cc/2022/Conference — NeurIPS 2022 Accept_

### Official Review · Reviewer_kQia · 2022-07-10

**Rating:** 5
**Confidence:** 3
**Soundness:** 3 good
**Presentation:** 2 fair
**Contribution:** 3 good

**Summary:**

This paper proposes to address the problem of end-to-end stochastic optimization. That is; finding argmin_a f(y, a) for some cost function f, input variables y, and actions a. The key caveat here is that y is not observed and instead, we observe x, a variable correlated with y (but the relationship between x, y is not provided).

Standard approaches to this problem learn a model of p(y|x), then use an off-the-shelf stochastic optimizer to find argmin_a E_{p(y|x)}[f(y, a)]. This approach of course can accrue errors due to the p(y|x) model not being perfect leading to subpar results.

This work proposes to learn the p(y|x) model, not to predict y, but such that the induced optimal action a performs best. To do this, they parametrize an energy-based model by taking the expected value of the cost function over the p(y|x) model, i.e. E(a; x) = E_{p(y|x)}[f(y, a)] and train this EBM to maximize likelihood of obtaining the ground truth optimal actions a, given the input x.

The model is trained using contrastive divergence but an importance sampling estimator is used to deal with the partition function.

Given that we also have knowledge of the true y for each x in the dataset, we also have access to a noiseless version of the loss function, not available at test time. This information is utilized by adding a regularizer to encourage low KL divergence. Between the noisy model and the noise-free model.

The authors present results on 3 stochastic optimization problems and show that on average their approach either outperforms the baselines or performs comparable to the strongest baselines while having a considerably lower training cost.

**Questions:**


After reading this paper, I am left with a few questions. First, did you explore alternative methods for EBM training such as contrastive divergence, PCD, noise contrastive estimation, or score matching? As I mentioned above, at the scale of the presented experiments, I think it is likely that a number of these techniques could be successful.

Next, given that your training procedure relies on self-normalized importance sampling, did you perform any verification that the SNIS estimator is accurate? One way to do this would be to show the effective sample size of the importance samples. This can also be used to help tune many parts of the sampler such as the proposal distribution.

**Limitations:**

The work discusses some of the limitations of EBM training and evaluation but does not spend much time talking about the specific drawbacks of the proposed method. I think the work would benefit from a limitations section.

**Strengths And Weaknesses:**

Strengths:

This is an interesting application of the theory of energy-based models. It is always nice to see work embracing the methods that have been developed to work with and train unnormalized models, rather than replacing them with normalized surrogates. The proposed method is simple and enables end-to-end optimization of the target lost function rather than a multi-stage approach. Unfortunately, I am not very familiar with this problem-space and the baselines, so I cannot speak much to the quality of the experimental evaluation.

Weaknesses:

I am somewhat concerned by the procedure used to train the models and I feel like the authors could provide more evidence to verify that the method is working as expected. Partition function estimation and training in energy-based models is notoriously difficult and, in my experience, importance sampling estimators only tend to work in *very* small scale settings. In larger scale settings, MCMC-based training such as PCD or CD tend to scale much better. I would expect to see at least 1 MCMC-based training procedure as a baseline. But, given that all of the experiments are in relatively low-dimensional problems, we are in a regime where alternative EBM training procedures may also such such as score-matching, noise contrastive estimation, or conditional noise contrastive estimation. I believe the work would be improved if some investigation was done into the method chosen to train the model.

Having said that, I would love to see a larger-scale application -- something maybe with a few hundred or thousand variables. EBM training has been successfully scaled to very high-dimensional data so I expect the method should work. Again, I am not very familiar with this space so I am not sure if any such benchmark problems exist, but if they do, that would be nice to include.

Besides this, I found the experimental details to be very lacking. There is no mention of the number of importance samples used in training and the scales of the importance weighting distribution. The neural networks used in the model are not described very well and I had to re-read a few times to understand that p(y|x) = N(y; neural_net1(x), neural_net2(x)). Is that right? The types of neural networks should be explained.

There is also no discussion of how long the models are trained, how they are optimized, and how model selection is performed. This is very difficult for EBMs, and therefore people typically use proxies for model fit instead of test-set likelihood. For that reason, I feel like it would be important for the authors to comment on this in the paper.

---

> ### Author Response · Authors · 2022-08-02
> **Response to Review 4, Part 1**
>
> **Did you explore alternative methods for EBM training?  I think it is likely that a number of these techniques could be successful. I would love to see a larger-scale application -- something maybe with a few hundred or thousand variables. EBM training has been successfully scaled to very high-dimensional data so I expect the method should work.**
>
> Our main contribution is addressing end-to-end stochastic optimization using
> EBMs, instead of developing a more effective algorithm for training EBMs. We
> showed that such an EBM formulation addresses a number of key challenges in
> existing decision-focused learning (DFL).
>
> As our method is general, it can indeed use any EBM training algorithms
> and will benefit from advances in EBM [1,2]. Besides importance sampling,
> we also tried Langevin Dynamics with Contrastive Divergence (LD-CD) for EBM training. We found that LD-CD brings
> little performance improvement but incurs longer training time on our tasks
> (Table 5 compares the two on the power generator scheduling dataset). This
> phenomenon is likely because the optimization problems are relatively
> low-dimensional, thus importance sampling works well and enjoys better
> efficiency in this regime.
>
> We share the same view with the reviewer that the proposed method is expected
> to work well in high-dimensional space, given that EBM training has been
> successfully scaled to very high-dimensional data. In high-dimensional space,
> we also believe MCMC-based EBM training methods should work better, and they
> can be easily plugged into our framework for stochastic optimization. In the
> high-dimensional regime, our method should yield even larger efficiency improvements than
> decision-focused learning (DFL) baselines. This is because DFL needs to solve
> the stochastic optimization problem during each training iteration which has
> time complexity $O(D^3)$, while our method has time complexity $O(D^2)$. The DFL baselines will thus require extremely long training time for high dimensional data. Unfortunately, there are currently no
> public datasets for high-dimensional end-to-end stochastic optimization
> problems to our knowledge. We leave the development of high-dimensional
> datasets and benchmarks for stochastic optimization as future work.
>
> | Method     | Task loss | Training time (seconds/epoch) |
> | ----------- | ----------- |  -----------  |
> | Two-Stage      | 4.52       | 0.01|
> | DFL-QPTH   | 4.11      |93.12 |
> | SO-EBM with importance sampling   | 3.83        |0.68 |
> | SO-EBM with Langevin dynamics   | 3.86     |1.47 |
>
> Table 5. Comparison of importance sampling and Langevin dynamics for training
> SO-EBM on power generator scheduling.
>
> **Verification of self-normalized importance sampling. One way to do this would be to show the effective sample size of the importance sampler.**
>
>
> The effectiveness of the importance sampler can be verified by the following:
>
> 1) As shown in Table 5, the importance sampler (IS) has achieved almost the same performance with long-run Langevin dynamics, which is a strong evidence that the IS-based estimator is accurate.
>
> 2) We compute the effective sample size (ESS) when we draw 512 samples from the proposal distribution for the power scheduling problem. As we can see from Table 6, as the training progresses, the effective sample size increases. This is because we use the location of the expert action as the mode of the proposal distribution. Maximizing the probability of the expert action makes the mode of the model's distribution more aligned with the mode of the proposal distribution. Therefore, the effective sample size increases during training. This is another evidence that our importance sampler and the training objective are effective.
>
>
> | Training epoch     | 1 | 10 | 30 | 50|
> | ---| ---  |---  |---  |---  |
> | Effective sample size   | 21.2  |37.3 |50.1 |48.6 |
>
> Table 6. Effective samples with respect to the training epoch on power scheduling problem. We draw 512 samples from the proposal distribution.

---

> > ### Author Response · Authors · 2022-08-02
> > **Response to Review 4, Part 2**
> >
> > **There is also no discussion of how long the models are trained, how they are optimized, and how the model selection is performed. This is very difficult for EBMs, and therefore people typically use proxies for model fit instead of test-set likelihood.  For that reason, I feel like it would be important for the authors to comment on this in the paper.**
> >
> > Different from traditional EBMs in image generation which may need various training tricks, our SO-EBM is much easier to train. It is because 1) Image generation is a density estimation problem in a high dimensional space which is hard to evaluate. This is why people resort to proxies of test-set likelihood for the image generation task.
> > In contrast, our problem has an explicit evaluation metric -- task loss, and thus is much easier to do model selection. 2) The density of the image datasets is usually much more complicated than the landscape of the stochastic optimization function. Image distribution could have a large number of modes in a very high dimensional space. 3) Image generation needs to use much more complicated network architectures which are harder to train.
> >
> >
> >
> > For the running time, we already reported the training time/epoch in our results section, and all the methods are trained by the same number of epochs (detailed in Section 4 of the supplementary). For model optimization, all the methods use Adam. For model selection, the task loss on the validation set is used for all the methods.
> >
> > **I found the experimental details to be very lacking. There is no mention of the number of importance samples used in training and the scales of the importance weighting distribution. The neural networks used in the model are not described very well and I had to re-read a few times to understand that p(y|x) = N(y; neural_net1(x), neural_net2(x)). Is that right? The types of neural networks should be explained.**
> >
> > We did report the number of importance samples in Section 4 of the
> > supplementary. For all the experiments, we use 512 samples. We will also add
> > the scales of the importance weighting distribution in the supplementary.
> >
> > Yes, p(y|x) = N(y; neural_net1(x), neural_net2(x)), where neural_net1 and
> > neural_net2 share the same layers except for the last layer. This is a common
> > architecture in heteroskedastic regression. We will add more descriptions of
> > the network architecture in the supplementary.
> >
> > ## References:
> >
> > [1] Yilun Du, Shuang Li, Joshua Tenenbaum, Igor Mordatch, Improved Contrastive Divergence Training of Energy Based Models, *ICML 2021*.
> >
> > [2] Erik Nijkamp et al, MCMC Should Mix: Learning Energy-Based Model with Neural Transport Latent Space MCMC
> > *ICLR 2022*

---

> > > ### Comment · Reviewer_kQia · 2022-08-06
> > > **Thanks**
> > >
> > > I thank the authors for their detailed response to my comments. Regarding the training methods, I am glad that the authors added a comparison with CD training and that the method performs similarly. I understand that there may not be available datasets for this problem in higher dimensions, which is not their fault, but the proposed training method will likely not scale when such data becomes available. CD training scales to high-dim data whereas IS does not. Its good to know that on this scale, CD performs as well as IS, indicating a pathway to scale this method. I further appreciate the ESS numbers for the IS. In my work, I've found IS methods can be deceptive and can appear to work when in reality the sampler isn't doing much. These ESS numbers look good to mean and give me more faith that the model is being trained well.
> > >
> > > Based on this, and the detailed responses to the other author's comments, I will raise my score to favor acceptance. Thank you.

---

### Official Review · Reviewer_pnpr · 2022-07-11

**Rating:** 6
**Confidence:** 4
**Soundness:** 3 good
**Presentation:** 2 fair
**Contribution:** 3 good

**Summary:**

The authors propose an approach for "decision-focused learning" (DFL).  This is where there is a prediction stage to predict stochastic parameters from some input features, that are then used to solve an optimization problem to determine an optimal action.  For example, an optimization of electricity generator scheduling based on forecasted electricity load / demand.  The old two-stage approach would be to separately fit a forecast model, and then use its predictions in an optimization problem solved by optimization solvers.  The DFL approach is to fit the forecast model in conjunction with the optimization problem, so the forecasts are optimal for the down-stream optimization problems themselves.

To do this for modern models using neural networks for the forecasters would generally require computing and back-propagating the gradient through the optimization loss and problem to the model parameters.  However, the gradient of the optimal action with respect to the forecast / prediction values is not always differentiable and the optimal action is the solution to an optimization problem, so computing gradient updates can be challenging and time consuming.

Most past work tended to either make restrictive assumptions or approximations, or used computational complex and inefficient implicitly differentiable layers.   This work proposes a workaround by modeling the optimal actions / optimization problem solution as a function of the features directly, but ties it to the forecast model as the energy term of the energy model is given by the expected optimization loss of under the forecast model (the forecast gives the distribution over the predicted parameters for the optimization problem).  In this way the energy model can be trained directly on historical examples with optimal solutions previously derived, by minimizing the negative log likelihood (which in turn optimizes the forecast model for the optimization problem) along with a KL-divergence regularizer added to the objective as well.

The authors compare their proposed approach to past methods on 3 datasets / tasks from different domains, and show it consistently has as good or better performance to other past DFL approaches, while also having as good or better run time.

**Questions:**

Please see questions and suggestions raised under weaknesses.

It would be good to mention and contrast with the other related work mentioned above.

How does the proposed approach handle constraints?  What is the impact of not including constraints?  What is the impact on large numbers of constraints / more complex optimization problems?

How much data does the proposed approach require?  How does the amount of data affect performance?

-Figures 4, 5, and 6 - labels are not lined up with the bars so it's hard to match the methods to the bars

**Limitations:**

There doesn't seem to be discussion of what are the limitations of this approach - it could benefit from some thought and discussion around what the limitations are.  Perhaps this would include dealing with constraints, or it could be the case where a large number of solved problems are needed for it to work well, as this should also be investigated ideally.

**Strengths And Weaknesses:**

Strengths:
-The authors point out key limitations of past work and aim to tackle a significant and important problem.  I.e., most practical applications of ML involve combining predictions and optimization and the DFL approach is arguably the way things should be done for all these applications / ML in general.

-The authors propose a novel solution that is interesting and shows promising results.

-The authors provide experimental results on a variety of different tasks and compare to multiple past DFL methods.

-The authors pledge to make the code publicly available, which will help a lot with reproducibility and understanding the approach.


Weaknesses:

-One piece of the problem, in particular the optimization part, that seems overlooked and not mentioned is the constraints.  Constraints are not included at all in the model of the end-to-end problem (mapping the features to the action) - they could only be included implicitly since the optimal actions, a, the model is trained on must satisfy the constraints.  It would help to have discussion about why it's not necessary to consider the constraints and what kind of impact it would have.  My speculation is that this can actually have a negative impact, because it puts more burden on the forecast neural network part of the model to account for the constraints (this model then needs to be more complex) - as what naturally fits the data better would have to be abandoned to predict a solution that leads to the best action when dropping the constraints.  I feel the problems chosen had very few and basic constraints which may be why the approach still worked in the experiments, but I would be curious to see how more constraints, and more complex constraints, might impact the performance.  It would also be useful to see how the learned forecast model compares to the separately trained model, and what cases it's making predictions farther off from the true values.

-There is one closely-related work overlooked here that appeared in the AAAI-22 Workshop on AI for Decision Optimization - this pointed out the same limitations and similarly targeted a somewhat similar approach.  However, that approach was different in that it approximated the optimization algorithm itself with a neural net, so the solution could be easily computed and differentiated, as opposed to effectively approximating the entire end-to-end problem as was done here (though here essentially only using the neural net to approximate the forecast piece of the model).  It would be good to mention this alternative approach, and it would be great if the two could be compared, as I'm very curious to see how they compare.  Paper:
"End-to-End Learning via Constraint-Enforcing Approximators for Linear Programs with Applications to Supply Chains"
https://research.ibm.com/haifa/Workshops/AAAI-22-AI4DO/PDF/End-to-End%20Learning%20via%20Constraint-Enforcing%20Approximators%20for%20LinearPrograms%20with%20Applications%20to%20Supply%20Chains.pdf

-The clarity of the presentation and description could be improved.  It's hard to follow the details of the method and the experiments.  For instance, the method description is not easy to follow, especially the KL-divergence regularization and the rationale behind it.  Another example, the experiment procedure is not clearly explained - i.e., for all methods, is the model fit and then just the forecast part used in conjunction with an optimization solver to get the final action / decision?  If so, what solver is used?  How are various hyper parameters set for different methods?

-Further experimental study would be useful.  In particular it would be good to see the impact on training data size on the performance of the different approaches.  In particular, with enough training data one would not expect the DFL approach to out-perform the two-stage one.  I also wonder if this approach may need more training data points to work well as it directly models the action given the inputs.
 Additionally run times will also be affected by amount of training data.  It would be good to understand these trade-offs.  Additionally it would be best to see the impact of varying lambda.   Additionally, what is the impact of using different network architectures / sizes?

---

> ### Author Response · Authors · 2022-08-02
> **Response to Review 3, Part 1**
>
> **How does the proposed approach handle constraints? What is the impact of not including constraints? What is the impact on large numbers of constraints / more complex optimization problems?**
>
> Our method handles the constraints implicitly through the pre-processing step
> and explicitly through the inference step. In the preprocessing step, the
> expert actions are obtained by solving the constrained optimization problem.
> Our target is to maximize the probability of these expert actions using the
> energy-based parameterization. In the inference step, we solve the original
> constrained optimization problem again to ensure feasibility of the final
> actions.
>
> We also tried imposing the constraints on EBM training by drawing samples from
> the constrained space. This is achieved by drawing samples from the proposal
> distribution and then projecting onto the feasible space to estimate the
> partition function. However, we found this constrained EBM training
> unnecessary as the constraints are imposed explicitly at inference time. It
> brings little performance improvement at the cost of extra computation.
>
> Our method should work when the number of constraints increases. This is
> because 1) maximizing the probability of the expert actions lets the mode of
> the energy function align with the feasible space; 2) the solver used in the
> inference step guarantees the final action satisfies the constraints. That
> said, when the feasible space is extremely small, training the constrained EBM
> may have more benefits. To train EBMs in the constrained space, one direction
> is to project samples into the feasibility space, as we have explored earlier.
> Another direction is to explore adding soft constraints to the energy function
> during training, *e.g*., Augmented Lagrangian penalty, Barrier penalty, and
> Posterior regularization. Such soft constraints can be easily incorporated
> into our model training.
>
> **It would also be useful to see how the learned forecast model compares to the separately trained model, and what cases it's making predictions farther off from the true values.**
>
> Table 1 provides a case study of the forecasted electricity demands of different
> methods for the power scheduling problem. In this example, the prediction of
> the two-stage model is closer to the ground truth, while SO-EBM tends to
> over-estimate the true demands. This is because the under-generation penalty
> coefficient is much larger than the over-generation penalty coefficient in this task.
> Therefore, to achieve better decision making loss, our model learns to be
> more cautious in under-estimating the demands. It learns the rule that
> under-generation leads to much larger penalty than over-generation. This also
> confirms the motivation of end-to-end stochastic programming -- smaller
> prediction loss does not always translate to better task loss, and it is
> necessary to tailor the learning of the predictive model to the downstream
> optimization task.
>
> | Method   | Forecasted demand|
> |------|------|
> | Ground Truth | 1.58 |
> |  Two-stage | 1.55| |
> | SO-EBM   | 1.63| |
>
> Table 1. Case study of the forecasted electricity demands of different methods.
>
> **It would be good to mention and contrast with the other related work mentioned above.**
>
> Thanks for suggesting this insightful paper and we will definitely cite it in
> the final version. This paper proposes ProjectNet to avoid solving the
> original optimization problem and improves the training efficiency of DFL. Our
> method is different from ProjectNet: (1) ProjectNet approximately solves the
> linear problems using a differentiable projection architecture, while our
> method directly formulates the problem using the energy-based model and
> maximizes the probability of the expert actions during training. (2)
> ProjectNet can only be applied to linear programming, while our method is more
> general and can be applied to any optimization problem. Applying ProjectNet to our studied problems is *non-trivial*, as it is designed for linear programming but all the problems in our experiments are non-linear.

---

> > ### Author Response · Authors · 2022-08-02
> > **Response to Review 3, Part 2**
> >
> > **How much data does the proposed approach require? How does the amount of data affect performance? In particular, with enough training data one would not expect the DFL approach to out-perform the two-stage one.**
> >
> > Table 2 provides the task losses under different ratios of training data on
> > the power scheduling problem. As we can see, our method outperforms the
> > baselines constantly except for the ratio of $5\%$ percent. When the ratio is below
> > $5\%$ percent, all the methods cannot work properly due to the extremely low resource.
> > The superior performance of our method is because we design the energy
> > function as the expected task loss, which leverages the algorithmic structure
> > inherent in the optimization problem. Hence, our method is not very data
> > demanding. Table 3 shows the training time/epoch under different ratios of
> > training data. As we can see, our method reduces the training time
> > significantly compared with DFL-QPTH across different amounts of training
> > data.
> >
> > Having enough training data to obtain a near-perfect predictor is rarely the case in real-world applications. Even with infinite data, the two-stage method will not necessarily outperform the DFL approach. This is because the model class of the predictor typically cannot perfectly match the true underlying distribution. For example, in the regression problem, we usually assume the target distribution follows a Gaussian distribution for computational tractability. However, it is usually incorrect for real-world data and thus a perfect predictor is unavailable even with enough training data.
> >
> > | Ratio of data (percent)   |$5\%$   | $10\%$ | $20\%$ | $50\%$| $100\%$|
> > | ----------- | ----------- |  -----------  |  -----------  |-----------  |-----------  |
> > | Two-Stage      | 5.99       |5.60 | 5.48 | 5.00|4.52|
> > | DFL-QPTH   | 5.96      | 5.29| 5.09| 4.67|4.11|
> > | SO-EBM   | 6.02       | 5.14| 5.06 |4.26 | 3.83|
> >
> > Table 2. Task losses under different ratios of training data on power scheduling problem. The full training data size is
> >
> > | Ratio of data (percent)   |$5\%$   | $10\%$ | $20\%$ | $50\%$| $100\%$|
> > | ----------- | ----------- |  -----------  |  -----------  |-----------  |-----------  |
> > | Two-Stage | 0.0051 |0.0059  |0.0065 |0.0071 |0.010|
> > | DFL-QPTH   | 4.46 | 10.70| 28.20|40.32 |93.12|
> > | SO-EBM   | 0.059| 0.078| 0.14|0.25 |0.68|
> >
> > Table 3. Training time (seconds/epoch) under different ratios of training data on power scheduling problem.
> >
> > **It would be best to see the impact of varying lambda. Additionally, what is the impact of using different network architectures/sizes?**
> >
> > We provide the impact of varying $\lambda$ for the power scheduling problem in Table 4. As we can see, when $\lambda$ is too small or large, the performance decreases. This is because when $\lambda$ is too large, the model focus more on estimating the entire loss landscape and the ability to capture the optimal point decreases. Therefore, the performance degrades. When
> > $\lambda$ is too small, the model only maximizes a single data point $a^{*}$ for each conditional distribution $q(a|\mathbf{x};\theta)$ and ignores the overall landscape of the optimization loss. Optimizing a single point for each conditional distribution easily causes overfitting and thus the performance decreases.
> >
> > | $\lambda$     | 0 | 0.1 | 0.5 | 1 | 1.5| 2 | 3|
> > |---|---|---|---|---|---|--|--|
> > | Task loss| 3.89 |3.89|3.86|3.83| 3.83|3.90|4.01
> >
> > Table 4. Impact of varying $\lambda$ for the power scheduling problem.

---

> > > ### Author Response · Authors · 2022-08-02
> > > **Response to Review 3, Part 3**
> > >
> > > **The clarity of the presentation could be improved. For example, it is not easy to follow the rationale behind the KL-divergence regularization.  Another example, the experiment procedure is not clearly explained - i.e., for all methods, is the model fit and then just the forecast part used in conjunction with an optimization solver to get the final action/decision? If so, what solver is used? How are various hyper-parameters set for different methods?**
> > >
> > > We will update the presentation to make it more clear in our final version. The KL-divergence regularization helps better learn the overall landscape of the optimization problem and hence reduces over-fitting. We have only single data points $a^{*}$ for each conditional distribution $q(a|\mathbf{x};\theta)$. Simply maximizing the probability of the single point ignores the overall matchness between the landscapes of the original optimization problem and the EBM-based probability density, which can cause over-fitting. The distribution-based regularizer essentially adds more anchor points besides the optimal decisions and thus helps fit the overall energy shape.
> > >
> > > Both the pre-preprocessing and forecasting parts use an optimization solver. The pre-processing step uses the solver to obtain expert actions on which the model is fitted. During inference, the model is used in conjunction with the solver to get the final actions.  To make the comparison fair, we use the same solver as the decision-focused learning baseline. Specifically, for the power scheduling and network security game problems, we use sequential-quadratic programming (SQP) based on the QPTH library [1,2]. For the COVID-19 resource allocation problem, we use the Cvxpylayer library [3].
> > >
> > > We use the same network architecture/size, optimizer, and number of training epochs for all the methods. The learning rate is selected from the same range based on the task loss on the validation set for all the methods. The hyper-parameters used in the experiments are included in Section 4 of the supplementary.
> > >
> > > **Figures 4, 5, and 6 - labels are not lined up with the bars so it's hard to match the methods to the bars**
> > >
> > > Thanks for pointing out this. We will make the labels of the figures more clear in our final version.
> > >
> > > ## References:
> > >
> > > [1] https://locuslab.github.io/qpth/
> > >
> > > [2] https://github.com/locuslab/e2e-model-learning/blob/master/power_sched/model_classes.py
> > >
> > > [3] https://github.com/cvxgrp/cvxpylayers

---

### Official Review · Reviewer_zDEe · 2022-07-12

**Rating:** 6
**Confidence:** 4
**Soundness:** 3 good
**Presentation:** 3 good
**Contribution:** 2 fair

**Summary:**

This paper tackles stochastic optimization problems of the form $\max_{a \in C} f(y, a)$, where $y$ is a stochastic variable and $a$ is an action taken by the learner given an observation $x$.

Recent work have introduced solutions that involve Deep Neural Networks. One body of work uses DNNs to make a prediction $\hat{y}$ given $x$, and then running an off-the-shelf optimizer for the inferred problem $f(\hat{y}, \cdot)$. This can lead to suboptimal solutions if $\hat{y}$ is not accurate. Another body of work can overcome this in certain scenarios by optimizing $(\hat{y}, a)$ jointly. They do so via end-to-end architectures that use so-called optimization-layers. These layers run a differentiable convex optimizer, and hence the incurred cost $f(\hat{y}, \hat{a} )$ can be optimized by jointly through backpropagation. A limitation of this approach is that the optimization layers must backpropagate through a convex optimiser; hence if it does not have a closed form solution, the computational cost can be steep.

The authors propose a novel architecture for joint optimization of $(\hat{y}, \hat{a})$ that avoids introducing optimization layers by taking an EBM approach. They construct an EBM that takes the expected cost $f(\hat{y}, \hat{a})$ under the model as the energy. To optimize the energy function, the authors operate a two-stage process. First, given a dataset $D = \\{ (x_i, y_i) \\}\_{i=1}^K$, they compute expert actions $a^i$ for each $y^i$ to construct an auxiliary dataset $D' = \\{ (x_i, a_i) \\}\_{i=1}^K$. They then use MLE to optimise the energy model under $D'$.

However, a main complication of this approach is that it requires stochastic outputs from the DNN. To this end, the authors rely on a Gaussian parameterization and self-normalized importance sampling. This means that the main computational difference between optimization-layer approaches and the author's approach is that they rely on stochastic relaxation instead of exact optimization.

The authors provide three synthetic experiments to evaluate their architecture: load balancing in power systems, healthcare resource allocation, and network security. In all cases, they show that their proposed method is competitive in terms of final performance while being computationally more efficient than optimization-layer based approaches.

**Questions:**

- The ability to generate expert actions is critical for this method to work. The authors suggest that previous method are limited in that they rely on convex solver - but it seems the proposed method suffer from the same limitation in that the solver is required to obtain expert actions?

- The authors opted for an EBM approach. Another equally valid method would have been expert distillation, which would have the benefit of not requiring estimating the energy function. Could the authors comment on why they opted against this simpler approach?

**Limitations:**

The authors discuss some limitations of the EBM approach, mainly the risk of overfitting. They provide an ablation to study the severity of the problem. The authors do not discuss other limitations, sensitivity to the choice of sampler and sensitivity to the choice of solver for expert actions.

**Strengths And Weaknesses:**

Strengths
- The paper is generally well written. The main idea, its motivation, mechanics, and intuition for design choices are all clearly spelled out. The experiments are well described with uncertainty estimates.
- The proposed architecture is quite clean and follows naturally from the EBM perspective.
- The experiments are well executed; clearly described, with thorough analysis of results.

Weaknesses
- The main contribution stated on page 2 does not seem to be well supported. First, the authors claim that the method is end-to-end, but this is not true since to construct the dataset, an off-the-shelf solver is needed to generate expert actions. Second, it does require solving the optimization problem (though not backpropagating through the solver), and while the EBM objective does not require convexity, imputing expert actions does. Overall, I think the paper can be strengthen significantly by a more nuanced discussion of the relative merits of the proposed method.
- The experiments are somewhat underwhelming in that they are all synthetic. It would have been significantly more interesting to test the proposed architecture on a real world problem with well-developed benchmarks. Moreover, the results themselves suggest that the gains are marginal; in most cases, the two-stage approach of pretraining a predictor $\hat{y}$ is faster and the loss in performance is not statistically significant.

---

> ### Author Response · Authors · 2022-08-02
> **Response to Review 2**
>
> **The method is not end-to-end since to construct the dataset, an off-the-shelf solver is needed to generate expert actions.**
>
> The solver is only used in the *pre-processing* step, and our method is indeed
> end-to-end during learning. We use the term 'end-to-end' to emphasize that we
> jointly learn the predictor and optimizer for the downstream decision making
> task, instead of learning them separately. The term 'end-to-end' is to
> differentiate our method from two-stage methods where the learning of the
> predictor ignores the downstream stochastic optimization problem. We will make
> it more clear in our final version.
>
> **The method does require solving the optimization problem though not backpropagating through the solver. Therefore, it still relies on a convex solver.**
>
> We want to clarify that our method does not rely on a convex solver. Rather,
> *any* existing black box solvers (including non-convex solvers!) can be applied
> in our method. This is because our method does not require the solver to be
> differentiable since we only use the solver to obtain the expert actions as
> preprocessing. In contrast, decision-focused learning requires the solver to
> be differentiable to enable backpropagation, and to date, such differentiable
> solvers are only available for convex optimization problems [1,2].
>
> **All experiments are synthetic. It would have been significantly more interesting to test the proposed architecture on a real-world problem with well-developed benchmarks.**
>
> The first two experiments are actually based on real-world problems and data:
> (1) The power scheduling problem uses real-world PJM electricity system
> operator data [3]. And we use the well-developed forecasting model and problem
> formulation from [2], one of the most well-known papers in this area. (2) The
> hospital resource allocation problem uses COVID-19 hospitalization data from
> CDC [4] and is an important problem in public health. And we use the SEIR-HD
> model [5], which is an advanced model for capturing the dynamics of the
> COVID-19 pandemic.
>
> Our third experiment on network security game uses a synthetic dataset from a well-known paper [6] in this area. We choose this problem because it is non-convex and deviates from the first two problems. We believe the three problems we choose are both diverse and have high real-world impact.
>
> **Gains are marginal.**
>
> Our method's gains are not marginal compared with decision-focused learning
> (DFL) methods. For example, in the power scheduling problem, our method
> improves the task loss by 7.3 percent compared with DFL and 18.6 percent compared with
> the two-stage model. Furthermore, our method has significantly lower
> computational cost. In terms of efficiency, our method is **136** times
> faster than DFL for the power scheduling task.
>
> **How does the method compare with expert distillation?**
>
> We did try expert distillation, but its performance is limited. Expert distillation is similar to the baseline Policy-network, which direct maps from input features to decision variables using supervised learning. Though we have tuned its hyper-parameters extensively, it still cannot achieve good performance (2x-3x larger task loss than the two-stage baseline). This is because it is a pure end-to-end model that directly models actions from input features. Such a design ignores the algorithmic structure inherent in the optimization problem and is extremely data hungry. This also shows the necessity of estimating the loss landscape with our energy-based formulation.
>
> **Sensitivity to the choice of sampler and solver for expert actions.**
>
> Our method is not restricted or sensitive to the sampler and solver we chose
> for expert actions. For the sampler, our method is a general framework for
> end-to-end stochastic optimization using energy-based model (EBM). While we
> choose the importance sampler for EBM training due to its efficiency, it is
> easy to plug in any samplers or other methods for EBM training. For example,
> we have also tried the Langevin dynamics sampler for model training, which
> leads to similar performance but has higher computational cost.
>
> For the solver, we use the same solver as the decision-focused learning
> baseline for a fair comparison. As aforementioned, our method does not require
> the solver to be differentiable. Hence, it can readily use and benefit from
> any advanced optimization solvers.
>
> ## References:
>
> [1] Akshay Agrawal, et al, Differentiable Convex Optimization Layers, *NeurIPS 2019*.
>
> [2]  Priya L. Donti, Brandon Amos and J. Zico Kolter, Task-based End-to-end Model Learning in Stochastic Optimization, *NeurIPS 2017*.
>
> [3] https://dataminer2.pjm.com/list.
>
> [4] https://gis.cdc.gov/grasp/covidnet/covid19_5.html.
>
> [5] Morgan P Kain, et al, Chopping the tail: How preventing superspreading can help to maintain covid-19 control. *Epidemics 2021*.
>
> [6] Kai Wang, Bryan Wilder, Andrew Perrault and Milind Tambe, Automatically Learning Compact Quality-aware Surrogates for Optimization Problems, *NeurIPS 2020*.

---

### Official Review · Reviewer_gqgA · 2022-07-14

**Rating:** 8
**Confidence:** 4
**Soundness:** 4 excellent
**Presentation:** 4 excellent
**Contribution:** 4 excellent

**Summary:**

This paper proposes a method to mitigate some of the computational challenges associated with the decision-focused learning paradigm under stochastic optimization-based decisions with unknown parameters. In particular, the authors construct (a) an energy-based surrogate function for the stochastic optimization-based decisions, (b) a loss function encouraging both good predictive accuracy and good global behavior of the energy-based surrogate, and (c) an efficient training procedure based on a mixture-of-Gaussians proposal. This enables them to avoid expensive implicit KKT differentiation steps that are needed when representing the decision procedure exactly as an optimization problem, as well as better enabling the representation of non-convex decision-making processes. The authors provide extensive experiments on several settings, and show empirically that their method achieves comparable or improved performance over baseline methods while drastically improving on computation time.

**Questions:**

* Section 4.1: Why does SO-EBM improve over DFL-QPTH? It would be useful to give some intuition for this, as at first glance it seems strange that a method using a surrogate loss would outperform a method using a more “exact” version of the loss.
* What are some of the limitations and potential negative impacts of the method?

**Limitations:**

In the checklist, the authors mention that they describe limitations and potential negative societal impacts of their work, but I cannot find this discussion. This discussion should be made more explicit, and should acknowledge that while the applications presented in the paper are social good applications (which I appreciate), the method is general enough that it may also have negative uses.

**Strengths And Weaknesses:**

Strengths:
* The proposed method is well-motivated, nicely integrates concepts from energy-based learning into the decision-focused learning literature, and mitigates a number of existing challenges with decision-focused learning paradigms.
* The experimental evaluation is thorough and well-done, demonstrating performance on different kinds of settings with respect to the form of the task loss, and showing insightful ablations. The method strongly outperforms other baselines (and the authors both include error bars *and* transparently leave in one setting where a baseline outperforms their method, both of which I appreciate).
* Everything is very well-explained, making the paper an absolute pleasure to read.

Weaknesses:
* In the checklist, the authors mention that they describe limitations and potential negative societal impacts of their work, but I cannot find this discussion.

Questions:
* Section 4.1: Why does SO-EBM improve over DFL-QPTH? It would be useful to give some intuition for this, as at first glance it seems strange that a method using a surrogate loss would outperform a method using a more “exact” version of the loss.

Minor points:
* Section 2, Problem Formulation paragraph: The terms $M$ and $a^*(x;\theta)$ should be more formally defined in the text (not just in Figure 1), as the rest of the problem formulation and method hinge on this. In particular, the current statement about learning a model that “takes the features $x$ as input and outputs the optimal decisions $a^*(x;\theta)$” could be misunderstood as a model that maps directly from data to decisions, without the intermediate prediction of the distribution.
* Section 2, Problem Formulation paragraph: As previously defined, $\mathcal{D}$ is the dataset. However, the optimization loss is introduced as “the expected decision cost under the joint distribution.” The wording here should be clarified. The goal is to learn the expected decision cost under the joint test-time distribution, *as approximated by* the dataset at training time, which is what the loss shows.
* Line 92: The left side should be the total derivative, rather than the partial derivative. Similarly, $\partial y/\partial \theta$ should be $\mathrm{d}y/\mathrm{d}\theta$. Similarly, in Equation (6), some of the partial derivatives should be total derivatives.

Typos:
* Line 84: “DFL method” -> “DFL methods”
* Problem Formulation paragraph in Section 2: “optimal decisions minimizes” -> “optimal decisions minimize”
* Line 99: “CVXlayers” -> “cvxpylayers”
* Line 100: “grammer” -> “grammar”
* Line 132: The optimization problem is missing $\mathbf{x}_i$
* Line 305: “target node” -> “target nodes”

---

> ### Author Response · Authors · 2022-08-02
> **Response to Review 1**
>
> **Why does SO-EBM improve over DFL-QPTH? It is strange that a surrogate loss can outperform a more exact loss.**
>
> The loss of DFL-QPTH is not necessarily more exact than SO-EBM, because QPTH can  be directly applied only to quadratic optimization problems. For non-quadratic optimization problems, DFL-QPTH needs to use sequential quadratic programming (SQP) to iteratively obtain the solutions. Differentiating through all the steps of SQP is prohibitively expensive in memory and time. To address this issue, existing works differentiate through just the last step of SQP (see code in [1]) to obtain approximate gradients. The inaccurate gradients may accrue during training and thus hurt decision quality. We will add more illustrations in the final version.
>
> **What are some of the limitations and potential negative impacts of the method?**
>
> One limitation of our method is that when the evaluation of the optimization function is expensive, the time efficiency of our method will be reduced.
> This is because we also need to evaluate the task loss of the samples drawn from the importance sampler during training. This limitation is also shown in our third example where evaluation of the defender's utility requires a matrix inverse. Our method needs to evaluate the utility for all the samples drawn from the proposal distribution which results a larger matrix to inverse. Therefore, the training time of our method is similar with the surrogate baseline in this problem. We will add a  dedicated limitation section to further discuss the limitations and point out future research directions of our method.
>
> As for the negative impacts, we agree that while our method can be applied to general end-to-end stochastic optimization problems, this presents opportunities for potential misuse based on the task at hand. However, we also believe that our method opens up avenues for many social good decision-making problems as shown in our experiments.
>
>
> **Minor points and typos.**
>
> Thanks for the nice suggestions and pointing out the typos! We will update accordingly and fix the typos.
>
> ## References:
>
> [1] https://github.com/locuslab/e2e-model-learning/blob/master/power_sched/model_classes.py

---

> > ### Comment · Reviewer_gqgA · 2022-08-07
> > **Response**
> >
> > Thank you to the authors for their response! Very good point about the differentiation through the last step of SQP introducing some imprecision in the DFL-QPTH method, and I look forward to the authors illustrating this more explicitly.
> >
> > A minor point - in the response to Reviewer zDEe, the authors write:
> > > In contrast, decision-focused learning requires the solver to be differentiable to enable backpropagation, and to date, such differentiable solvers are only available for convex optimization problems
> >
> > Current decision-focused learning does not require the solver itself to be differentiable. In particular, an optimal solution can be obtained via any black-box solver, and then this optimal solution used for implicit differentiation (of, e.g., the KKT conditions) in the backpropagation step. Instead, the challenge is that for non-convex problems, the KKT conditions are not guaranteed to correctly characterize the gradients - so it is the backwards step (not the forwards step) that has required convexity when using implicit differentiation of KKT conditions.

---

> > > ### Author Response · Authors · 2022-08-08
> > > **Response to the minor point**
> > >
> > > Thanks for your responses! We agree that the solver itself does not need to be differentiable. We used ‘differentiable solver’ to refer to existing differentiable optimization libraries, e.g., QPTH and Cvxpylayers, that are able to backpropagate through the optimal solution of the solver. Such a differentiable solver library consists of a black-box solver and implicit differentiation (e.g., KKT conditions). Decision-focused learning can be problematic for non-convex problems since it needs to backpropagate through the optimal solution and the KKT conditions are not guaranteed to correctly characterize the gradients. In contrast, our method can mitigate this issue since we do not need to backpropagate through the optimal solution.

---

### Author Response · Authors · 2022-08-02
**Response to All Reviews**

We thank all the reviewers for their valuable comments. One common suggestion from
multiple reviewers is to add a dedicated limitation section. We will include such a
  section in the final version, and add more discussion on
the limitations of our method and future research directions.

---

### Meta-Review · Area_Chair_gHRX · 2022-09-04

**Recommendation:** Accept
**Confidence:** Certain

**Metareview:**

In agreement with all the reviewers, I recommend acceptance. In the final version, the authors should take into account the reviewers’ recommendations and update the paper accordingly. Also, regarding the suggestion of Reviewer kQia, I recommend that the author include a high-dimensional synthetic experiment to analyze how their method behaves in that scenario.

**Award:**

No

---

### Decision · Program_Chairs · 2022-09-14

Accept